# SARS-CoV-2 N protein recruits G3BP to double membrane vesicles to promote translation of viral mRNAs

Siwen Long[1], Mykhailo Guzyk[1], Laura Perez Vidakovics [1], Xiao Han[2,3], Renhua Sun [2], Megan Wang [1], Marc D. Panas [1], Egon Urgard[4], Jonathan M. Coquet [1,4], Andres Merits[5], Adnane Achour [2,3] & Gerald M. McInerney [1] ✉

Ras-GTPase-activating protein SH3-domain-binding proteins (G3BP) are critical for the formation of stress granules (SGs) through their RNA- and ribosome-binding properties. SARS-CoV-2 nucleocapsid (N) protein exhibits strong binding affinity for G3BP and inhibits infection-induced SG formation soon after infection. To study the impact of the G3BP-N interaction on viral replication and pathogenesis in detail, we generated a mutant SARS-CoV-2 (RATA) that specifically lacks the G3BP-binding motif in the N protein. RATA triggers a stronger and more persistent SG response in infected cells, showing reduced replication across various cell lines, and greatly reduced pathogenesis in K18-hACE2 transgenic mice. At early times of infection, G3BP and WT N protein strongly colocalise with dsRNA and with non-structural protein 3 (nsp3), a component of the pore complex in double membrane vesicles (DMVs) from which nascent viral RNA emerges. Furthermore, G3BP-N complexes promote highly localized translation of viral mRNAs in the immediate vicinity of the DMVs and thus contribute to efficient viral gene expression and replication. In contrast, G3BP is absent from the DMVs in cells infected with RATA and translation of viral mRNAs is less efficient. This work provides a fuller understanding of the multifunctional roles of G3BP in SARS-CoV-2 infection.

The coronaviral replication cycle is a complex process encompassing viral entry, uncoating, translation, replication, assembly, and release[1], each stage likely influenced by a number of host proteins, both pro- and antiviral. Despite having a relatively large coding capacity, with many essential functions carried out by the viral non-structural and accessory proteins, the coronaviruses rely on a complex interplay with host factors to successfully infect and propagate within host cells[2]. These host factors play pivotal roles in determining the outcome of the infection, modulating the host immune response, and offering potential therapeutic targets. A fuller understanding of the viral replication cycle will depend on uncovering the functions of these host factors.

G3BP1/2 (G3BP) are homologous multifunctional RNA-binding proteins, with a critical role in the nucleation of SGs[3], dynamic mRNA-protein complexes formed within host cells in response to various stressors, including viral infections[4]. Many viruses target G3BP and

[1]Division of Virology and Immunology, Department of Microbiology, Tumor and Cell Biology, Karolinska Institutet, Stockholm, Sweden. [2]Department of Medicine Solna, Science for Life Laboratory, Karolinska Institute Solna, Solna, Sweden. [3]Division of Infectious Diseases, Karolinska University Hospital, Stockholm, Sweden. [4]Department of Immunology and Microbiology, Leo Foundation Skin Immunology Research Centre, University of Copenhagen, Copenhagen, Denmark. [5]Institute of Bioengineering, University of Tartu, Tartu, Estonia. ✉e-mail: gerald.mcinerney@ki.se

other SG proteins to inhibit the formation of SGs or to otherwise promote viral replication[5]. Some viruses target the intact protein for inclusion into their replication complexes suggesting that pro-viral role(s) of the protein are also likely[6–8]. The functions of G3BP in viral replication is probably best understood for the Old World alphaviruses[9–12], in which the protein is bound and recruited to RNA replication complexes by the viral non-structural protein (nsP)-3[7]. nsP3 contains two FGDF motifs in an unstructured region close to the C-terminus, that mediate high affinity binding to a hydrophobic groove on the surface of the NTF2-like (NTF2L) domain of G3BP[9,13]. This interaction likely occurs immediately upon nsP3 synthesis, and bivalent binding leads to the formation of chains of G3BP dimers, interlinked by flexible C-terminal domains of nsP3. This binding mode is important for maintaining high local concentrations of viral replication complexes to promote efficient viral gene expression and genome replication at a critical early stage in viral replication[10,13].

Early in the COVID-19 pandemic, SARS-CoV-2 protein-protein interaction studies revealed an interaction between G3BP and the nucleocapsid (N) protein[14–16]. The interaction is dependent on an ITFG-based motif at the N-terminus of N[17], which binds to the G3BP NTF2L in a manner reminiscent of the binding to Old-World alphavirus nsP3, suggesting a similar binding mode, later confirmed by X-ray crystallography[18]. Indeed, the interaction is important for inhibition of SG formation during infection[19–23], but potential other pro-viral effects of this interaction in infected cells have not been described.

Here, we show that the recruitment of G3BP to viral replication-transcription complexes (RTC) occurs early in SARS-CoV-2 infection in a manner which provides an accessory proviral effect of promoting the local translation of viral mRNAs.

## Results
### The kinetics of SG assembly and disassembly during SARS-CoV-2 infection
We first studied the kinetics of SGs during SARS-CoV-2 infection. We observed that SGs were induced at 6 hpi (even in cells where N protein is not yet detected) and 12 hpi but were disassembled later in infection as N protein levels increased (Fig. 1a–c). In contrast, we observed sustained activation of protein kinase R (PKR) and consequent phosphorylation of eIF2α during the infection (Fig. 1d). To determine whether the disassembly of SGs during SARS-CoV-2 infection represented a specific mechanism, we performed experiments with exogenous stressors as described previously[24]. Sodium arsenite (SA) induces SG assembly via eIF2α phosphorylation[25], whereas pateamine A (PAT) does so by inhibition of eukaryotic initiation factor 4 A (eIF4A)[26]. We found that SARS-CoV-2 infection could interfere with SGs induction by SA or by PAT in VeroE6 cells (Supplementary Fig. 1a and Fig. 1e, f), indicating that the mechanism of SG inhibition is downstream of eIF2α phosphorylation or eIF4A inhibition. The results in Fig. 1 were repeated in U2OS-ACE2 cells with similar results (Supplementary Fig. 1b–h). In infected U2OS-ACE2 cells, we also detected activation of PKR-like ER kinase (PERK), suggesting collaboration of this kinase in eIF2α phosphorylation (Supplementary Fig. 1e).

### SARS-CoV-2 N, in founder strain and variants, binds G3BP
N protein (419 amino acids) is encoded by the ninth ORF of SARS-CoV-2 genome and is composed of N-terminal RNA-binding (N-NTD) and C-terminal dimerization domains (N-CTD) that are flanked by three intrinsically disordered regions (IDRs)[27] (Fig. 2a). Previous research revealed that N interacts with G3BP[14] and the interaction can lead to SG disassembly upon overexpression[17,19–23]. Here, we confirm that N protein bound to G3BP1 and G3BP2 in cells infected with the SARS-CoV-2 founder strain (Fig. 2b). Others have shown that the N-IDR1 domain of N protein binds to G3BP[21], and specifically an ITFG (aa 14-17) motif in N-IDR1 was essential for their interactions[17,18,23]. This motif is highly conserved among most SARS-CoV-2 variants. However, a P13L

mutation proximal to the ITFG motif emerged in Omicron and XBB.1.5 variants (Fig. 2c). Despite this mutation, Omicron and XBB.1.5 N proteins maintained binding capability to G3BP, although exhibiting a marginal reduction in affinity (Fig. 2d). However, this slightly decreased binding had little consequence for SG inhibition, since cells infected with either variant, showed similar induction of SGs and were inhibited from forming SGs upon exogenous stress (Fig. 2e, f).

### A SARS-CoV-2 mutant lacking the interaction with G3BP, is defective in SG inhibition and is attenuated in multiple cell lines
The crystal structure of G3BP1-NTF2L domain in complex with a peptide derived from SARS-CoV-2 N protein (aa 1–25) revealed that the docking is mediated by hydrogen bonds, van der Waals contacts and electrostatic interactions[18]. The NTF2L features a long binding groove formed by two α-helices and two β-sheets, consisting of a 5.6 Å wide groove and a 3.5 Å narrow groove. The aromatic ring of N-F17 inserts into the aromatic cage at the center of NTF2L binding groove, stabilized by multiple π-stacking interactions. Meanwhile, the bulky hydrophobic side chain of N-I15 inserts into the small groove, coordinated by NTF2L residues L10, V11, and P6[18] (Fig. 3a). Interestingly, the dual groove-insertion mode of G3BP1and ΦxFG motif was also observed in complexes involving Caprin1/G3BP-NTF2L, and alphavirus nsP3/G3BP-NTF2L[13,28], highlighting the ΦxFG motif is a dominant factor for the G3BP1-NTF2L-mediated protein interactions. Additionally, research indicates that mutations at N-I15 and at N-F17 are required to fully abrogate the interaction between G3BP1 and the N protein[29]. To evaluate the role of G3BP-N protein interaction in the context of viral infection, we mutated both I15 and F17 to alanine within the infectious icDNA clone of SARS-CoV-2[30] (Fig. 3b and Supplementary Fig. 2a–c). The SARS-CoV-2 RATA virus, which failed to form a complex with N and G3BP, induced a more robust SG response and was significantly less able to inhibit SGs after exogenous stress (Fig. 3c–e). These findings indicate that N-WT disrupted SGs formation more potently by interacting with G3BP whereas N-RATA, which lacks G3BP binding, shows diminished suppression of SG formation. Similar results were obtained in U2OS-ACE2 cell lines (Supplementary Fig. 3a–d).

To further investigate the contribution made by the G3BP-N interaction on SARS-CoV-2 replication, we assessed the replication kinetics of SARS-CoV-2 WT virus and the RATA mutant in a selection of different cell lines. RATA produced a smaller plaque size (Fig. 3f) and replicated to significantly reduced titers compared to WT virus in all G3BP-expressing cells (Fig. 3g). Thus, the G3BP-N interaction augments SARS-CoV-2 replication in a variety of in vitro systems. These findings align with a recent study demonstrating that a single mutation at F17, which reduces the interaction between G3BP and the N protein, attenuates SARS-CoV-2[23]. Loss of G3BP1 binding did not elevate interferon levels in RATA-infected cells (Supplementary Fig. 3e), indicating that the attenuation of the RATA virus is independent of G3BP1-mediated interferon signaling pathways. Interestingly, both WT and RATA viruses demonstrate enhanced replication in G3BP-deficient cells, likely due to the absence of SG assembly, which typically impedes viral mRNA translation. In cells lacking G3BP, RATA replicates to titers closer to those of WT virus (Fig. 3g), suggesting that its attenuation is largely due to the lack of G3BP interaction. However, there remains a 4-fold reduction in RATA titers relative to WT in cells lacking G3BP, indicating that the attenuation is partly G3BP-independent. Since we know from previous work[17] that the RATA mutation is very specific for G3BP, we do not believe the G3BP-independent attenuation to be due to loss of another protein binding partner. Instead, it could be an effect of the sensitivity of the N protein IDRs to mutation[31].

### SARS-CoV-2 RATA is not pathogenic for K18-hACE2 mice and protects against subsequent infection with SARS-CoV-2 WT
To determine whether loss of G3BP binding also attenuates RATA replication in vivo, we took advantage of the commonly used K18-

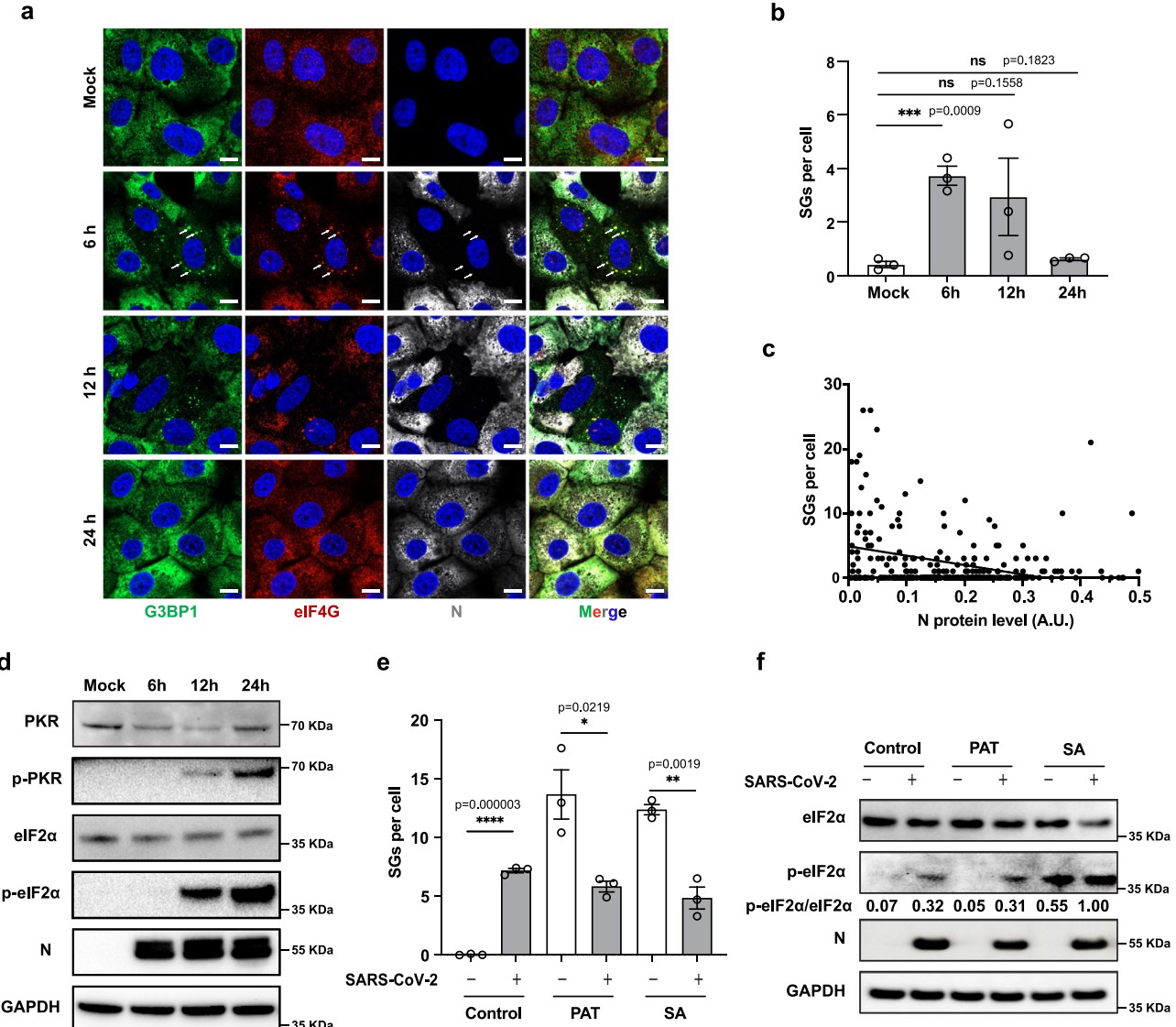

**Fig. 1 | SARS-CoV-2 infection inhibited SGs formation independent of eIF2α.**
**a** VeroE6 cells were mock infected or infected with SARS-CoV-2 WT at 0.5 MOI. Cells were fixed at 6, 12, and 24 hpi and stained for G3BP1 (green), eIF4G (red), N (gray), and Hoechst (blue). White arrows indicate SGs. Representative images from three independent experiments are shown. Scale bar = 10 μm. Quantification of SG foci and N protein intensity were analyzed by CellProfiler (**b**) Bars represent mean ± SEM for three independent experiments shown as hollow dots (each dot representing the mean of 30 cells). **c** The line indicates the inverse correlation of N protein level and SG numbers per cell ($n$ = 320 cells). The correlation was calculated by Pearson correlation coefficient $r$ = −0.2, $p$ = 0.0004 (two-tailed). **d** VeroE6 cells were infected as in (**a**). Cell lysates were separated by SDS-PAGE and probed with indicated antibodies. **e** VeroE6 cells were infected as in (**a**) and at 6 hpi were stressed with SA or PAT for 1 h before fixation and staining. Images are shown in Supplementary Fig. 1a. Quantification of SG foci was performed using CellProfiler. Bars represent mean ± SEM for three independent experiments, shown as hollow dots (each dot representing the mean of 30 cells). **f** Cells were lysed for immunoblotting with indicated antibodies.

hACE2 mouse model for severe SARS-CoV-2 pathogenesis[32]. Mice were intranasally inoculated with 100 PFU of either SARS-CoV-2 WT or RATA (Fig. 4a) and monitored for weight loss daily. As expected, K18-hACE2 mice infected with SARS-CoV-2 WT showed significant weight loss, starting at day 5 after inoculation and ending with sacrifice when 20% weight loss was reached (Fig. 4b). In contrast, mice infected with RATA did not display any detectable weight loss. Analyzes of viral RNA load in lung tissue revealed that RATA replication was significantly lower than WT (Fig. 4c). Histopathology of lungs of a selection of mice sacrificed at day 7 after infection revealed significantly higher amounts of immune cell infiltrates, and more extensive tissue damage in the lungs of mice infected with WT SARS-CoV-2 compared to mice infected with RATA (Fig. 4d). Since RATA-

infected mice displayed only mild disease, we evaluated whether infection with RATA protects mice from subsequent challenge with WT SARS-CoV-2. At day 22 after initial infection, we challenged mice that had been initially infected with RATA with 10x lethal dose (1000 PFU) of WT virus. Naive mice infected with 1000 PFU of WT virus, used as controls, lost weight rapidly and were all sacrificed on day 8 post infection or earlier. In contrast, mice that had previously been infected with the RATA mutant exhibited neither weight loss nor severe tissue damage (Fig. 4e–g). Altogether, these experiments demonstrate that RATA is significantly attenuated in the K18-hACE2 mice model and suggests the possibility that the inhibition of the G3BP-N interaction might be included in a strategy to construct a live-attenuated COVID-19 vaccine.

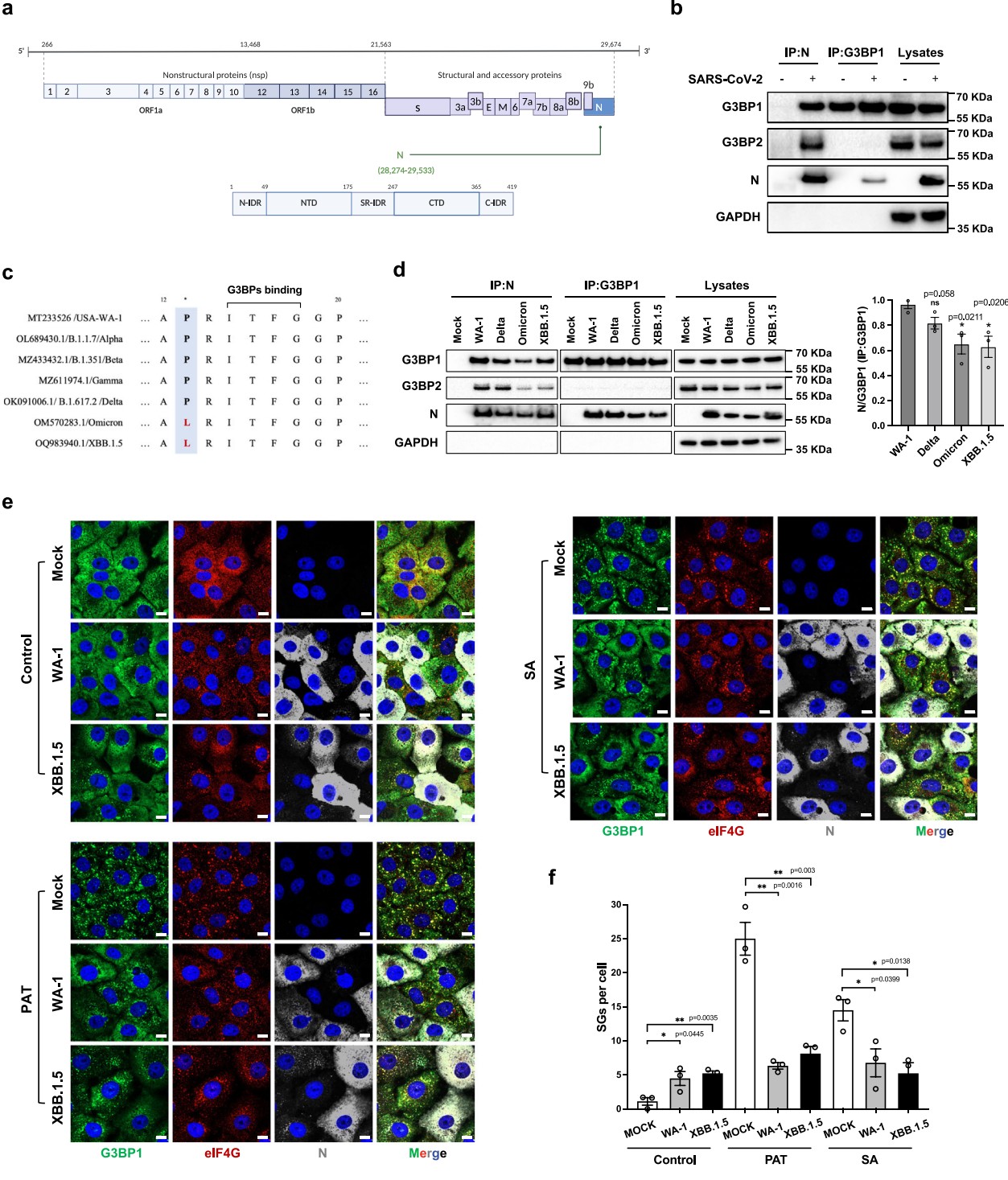

**Fig. 2 | SARS-CoV-2 N in founder stain and variants, binds G3BP. a** Schematic of SARS-CoV-2 genome and N protein, created in BioRender. **b** VeroE6 cells were mock infected or infected with SARS-CoV-2 WT (founder strain) at 0.01 MOI for 24 h. Cells were lysed and immunoprecipitated with G3BP1 or N antibodies and analyzed by immunoblot for indicated proteins. **c** Amino acid sequence of N (aa 12-20) in different variants. **d** VeroE6 cells were mock infected or infected with WA-1, Delta, Omicron or XBB.1.5 at 0.01 MOI for 24 h. Cells were lysed and immunoprecipitated using G3BP1 or N antibody to determine the interaction of G3BP and N.

Quantification of western blot was performed using Image J and data are presented as mean values ± SEM from three independent experiments. **e** VeroE6 cells were mock infected or infected with WA-1 or XBB.1.5 at 0.5 MOI. At 6 hpi, cells were stressed with SA or PAT for 1 h before fixation and staining with indicated antibodies. Representative images from three independent experiments are shown. Scale bar = 10 μm. **f** Quantification of SG foci was performed using CellProfiler. Bars represent mean ± SEM for three independent experiments, shown as hollow dots (each dot representing the mean of 30 cells).

## G3BP1 facilitates phase transition of N protein and G3BP-N interaction induces distinct lysates granules

N protein contains three IDRs and has been demonstrated to undergo liquid–liquid phase separation (LLPS) with the viral RNA genome, potentially facilitating viral RNA replication, transcription, particle assembly[33–36]. Previously, the I15A and F17A mutations in N-RATA showed no effect on assembly of virus-like particles[17]. To compare the LLPS properties of N-WT and N-RATA, we employed a technique allowing

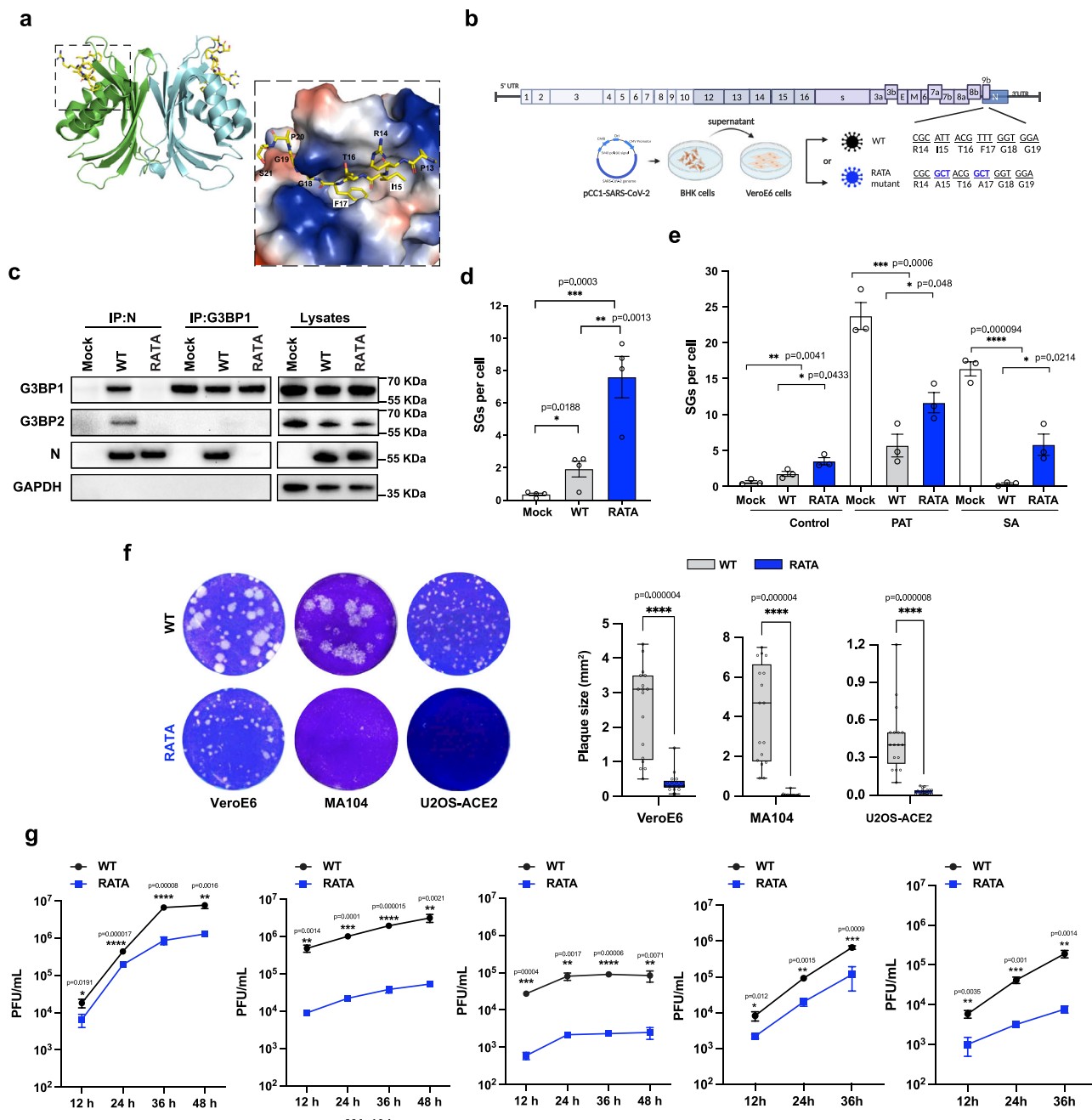

**Fig. 3 | SARS-CoV-2 RATA is defective in SG inhibition and is attenuated in multiple cell lines. a** Structure of G3BP1 NTF2L bound to N-WT (aa1-25) (PDB: 7SUO). **b** Schematic of SARS-CoV-2 WT and RATA mutant, created in BioRender. RNA sequencing confirmed no reversion at positions 15 or 17 in RATA clones, indicating successful recovery of infectious virus (Supplementary Fig. 2a, b) **c** VeroE6 cells were infected with WT virus or RATA mutant at 0.01 MOI for 24 h. Cells were lysed and immunoprecipitated with G3BP1 or N antibody for immunoblotting. **d** VeroE6 cells were infected with SARS-CoV-2 WT or RATA at 0.5 MOI for 6 h. Images are shown in Supplementary Fig. 2d. Quantification of SG foci was performed using CellProfiler. Bars represent mean ± SEM for four independent experiments shown as hollow dots (each dot representing the mean of 20 or 99 cells). **e** VeroE6 cells were infected as in (**d**) and stressed at 6 hpi with SA or PAT for

1 h. Images are shown in Supplementary Fig. 2e. Quantification of SG foci intensity was performed using CellProfiler. Bars represent mean ± SEM for three independent experiments shown as hollow dots (each dot representing the mean of 30 cells). **f** VeroE6 and MA104 – two SARS-CoV-2 permissive cell lines, and U2OS cells transduced with ACE2 and TMPRSS2 were infected with SARS-CoV-2 WT or RATA until plaques were visible. Representative images on left show relative plaque sizes in indicated cell lines. Right, plaque sizes were measured by Image J ($n = 17$ plaques per condition). The box plots bound the interquartile range divided by the median, with the whiskers extending from the minimum to the maximum values, each dot represents the plaque size of an individual plaque. **g** Viral titer from indicated cell lines infected with SARS-CoV-2 WT or SARS-CoV-2 RATA at 0.05 MOI. Data are presented as mean values ± SD as appropriate ($n = 3$ biological replicates).

high-fidelity reconstitution of ribonucleoprotein granules in a cell lysate-based system[37]. Notably, both N-WT and N-RATA exhibited droplets upon their individual addition to ΔΔGFP-G1-WT cell lysates (Fig. 5a), demonstrating that the N-RATA mutation does not affect the multivalent interactions with RNA and/or other N protein molecules. This

phenomenon was not affected by the lack of G3BP as both N-WT and N-RATA induced LLPS in the ΔΔGFP cell lysates (Supplementary Fig. 4a).

G3BP1 also exhibits the capability of LLPS attributed to its three IDRs[38]. To assess the impact of G3BP1 on the formation of N-induced droplets, we constructed a comprehensive diagram using increasing

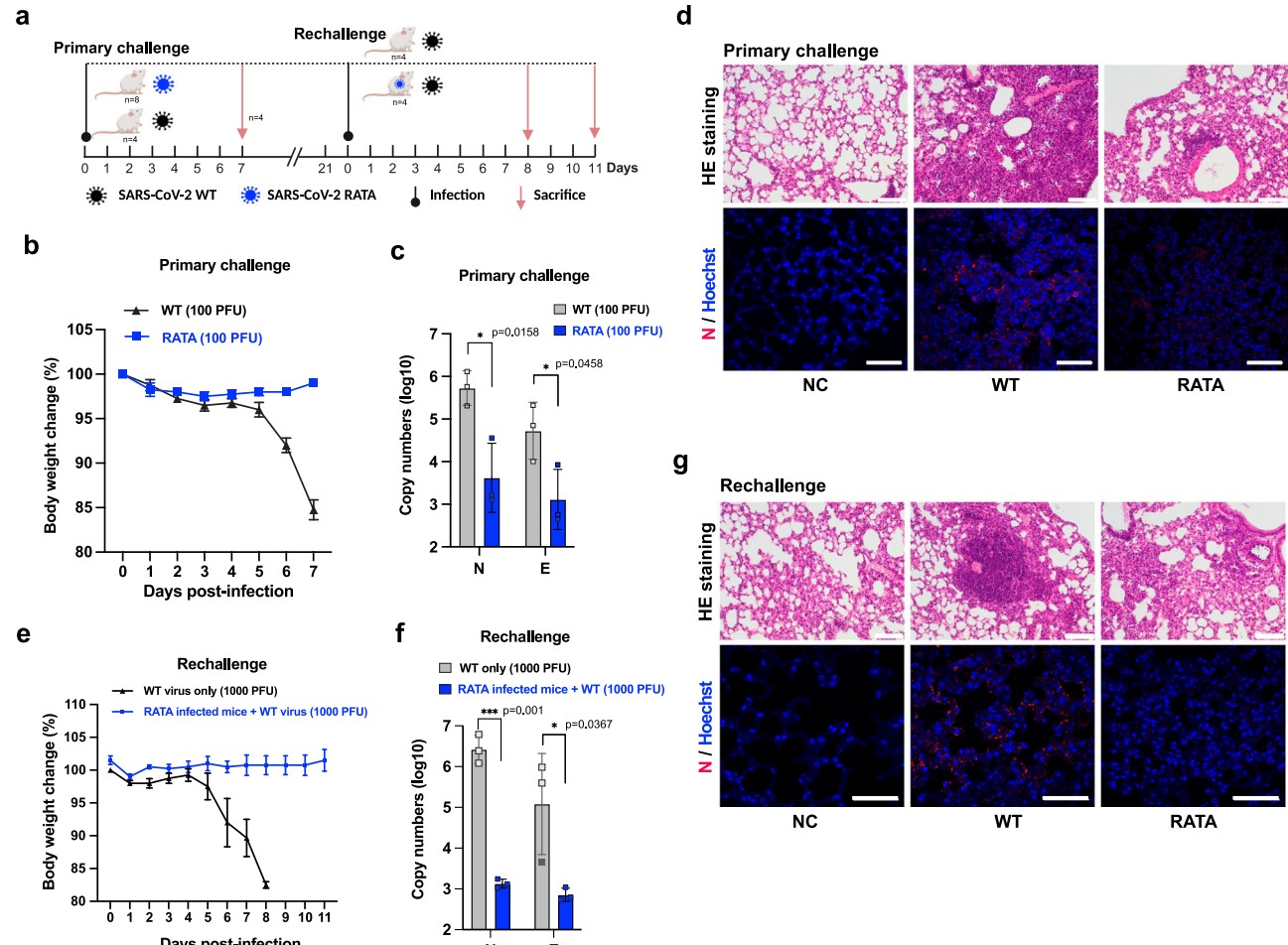

**Fig. 4 | Infection of K18-hACE2 transgenic mice with SARS-CoV-2 WT or RATA mutant. a** Schematic of SARS-CoV-2 challenge and rechallenge experiments, created in BioRender. **b** For primary challenge, mice were inoculated with 100 PFU of SARS-CoV-2 WT or RATA and evaluated for weight loss (*n* = 4 in each group). **c** RT-qPCR of N protein and E protein expression in mice lungs after primary challenge (*n* = 3 mice). Data are presented as mean values ± SEM as appropriate in (**b**, **c**). **d** Lung histopathology and N protein immunohistochemistry staining in the lungs at 7 dpi from mock, SARS-CoV-2 WT and RATA infected mice. Scale bar = 50 μm. **e** mice 21 days after the primary infection, or naive mice were rechallenged with 1000 PFU of SARS-CoV-2 WT and evaluated for weight loss (*n* = 4 in each group). **f** RT-qPCR of N protein and E protein expression in mice lungs after rechallenge (*n* = 3 mice). Data are presented as mean values ± SEM as appropriate in (**e**, **f**). **g** Lung histopathology and N protein immunohistochemistry staining from mock and rechallenged mice. Images are representative of lung sections from four mice, scale bar = 50 μm.

concentrations of purified G3BP1 and N proteins. We observed droplet formation for both N-WT and N-RATA proteins when the protein concentration reached 40 μM. Notably, below this threshold, G3BP1 robustly promoted the LLPS of N-WT in a dose-dependent manner, whereas the addition of G3BP1 to N-RATA resulted in only formation of aggregates (Fig. 5b and Supplementary Fig. 4b), demonstrating that G3BP1 enhances the LLPS of N protein by interacting with its ITFG motif.

As expected, G3BP1-induced droplets contained eIF4G and Caprin1, typical SG components but did not contain actin (Fig. 5c) and were inhibited from formation by addition of RNase (Supplementary Fig. 4c). However, in G3BP1-N droplets, there was no significant signal for eIF4G or Caprin1, revealing that the composition of N- containing droplets differs from SGs (Fig. 5c). Thus, the ability of N to undergo LLPS, combined with its high binding affinity for G3BP, outcompetes other proteins of the SG network, and that this competition might be the mechanism for the ability of N protein to disrupt SGs formation.

### N recruits G3BP to viral RTC at early times in infection via interaction with pore protein nsp3

N protein has been reported to dynamically localize to the viral RTC at the early stage of infection, contributing to efficient viral RNA replication and transcription[39]. In infected cells, we observed that while G3BP1 colocalized with N protein throughout the infection, G3BP1-N complexes were accumulated in close proximity to clusters of dsRNA-positive foci at 3 and 6 hpi, but not at 12 hpi (Supplementary Fig. 5a). Notably, Burke and colleagues previously showed that G3BP1 is excluded from sites of genomic RNA accumulation at later times in SARS-CoV-2 infection[40]. As expected, N-RATA did not colocalize with G3BP1 but did accumulate with dsRNA (Fig. 6a), demonstrating that SARS-CoV-2 N recruits G3BP1 to RTC early in infection.

Previous data indicated that the protein composition of G3BP1-N lysate granules differs from SG assemblies (Fig. 5c). To further distinguish the two different condensates and explore the mechanisms whereby the G3BP-N interaction contributes to SARS-CoV-2 replication, we used a U2OS cell line panel shown in Fig. 6b[3,10]. G3BP1 and eIF4A co-staining can be used to identify SGs[24], while dsRNA signal indicates RTC. Consistent with Fig. 6a, G3BP1 colocalized with N-WT and dsRNA in SARS-CoV-2 WT-infected ΔΔGFP-G1-WT cells. However, when those cells were infected with RATA, G3BP1 only clustered in SGs but not at RTC (Fig. 6c). Interestingly, nonstructural protein 3 (nsp3), a large transmembrane protein at the RTC pore[41], also clustered with G3BP1/N puncta (Fig. 6e), and co-precipitated with G3BP1 and N in

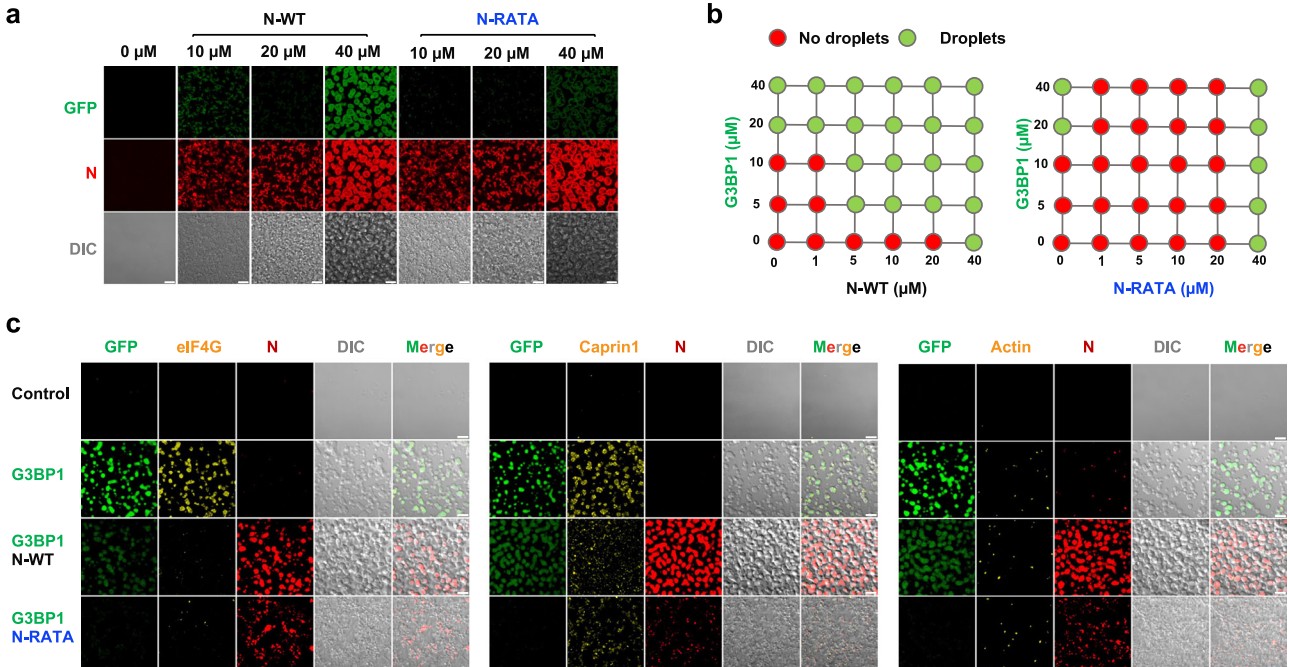

**Fig. 5 | G3BP1 facilitates LLPS of N and G3BP-N induces distinct lysate granules.**
**a** Purified N-WT or N-RATA protein was added at varying concentrations to lysates from U2OS cells lacking both G3BP1 and G3BP2, and stably expressing GFP-G3BP1 (ΔΔGFP-G1-WT). N-specific antibodies conjugated to Alexa Fluor 647 secondary antibody were used to visualize N protein. Representative images from three independent experiments are shown. **b** Summary of the phase separation behaviors of the N-WT or N-RATA with increasing concentrations of purified G3BP1 protein in ΔΔGFP-G1-WT cell lysate. Corresponding images are shown in Supplementary Fig. 4b. **c** Purified G3BP1 (20 μM) with or without N-WT (10 μM) or N-RATA (10 μM) to was added to ΔΔGFP-G1-WT cell lysates. Fluor-conjugated antibodies were used to visualize N (red), eIF4G/Actin/Caprin1 (yellow), or GFP (green) indicated GFP-G3BP1. Representative images from three independent experiments are shown. Scale bar = 10 μm.

SARS-CoV-2 WT-infected cells (Fig. 6f and Supplementary Fig. 5e), suggesting that G3BP1-N complexes are localized to the DMV pores, from which nascent viral mRNA emerges. These observations uncovered the significance of the G3BP-N interaction for SG disassembly and recruitment of G3BP1 to RTC. The removal of the RGG domain in G3BP did not affect its interactions at the NTF2L domain but led to an inability to rescue SGs formation under different stress conditions[3]. As a consequence, the dispersed eIF4A signals indicated the inability of G3BP1-ΔRGG to support SGs assembly during both WT virus and RATA mutant infections. However, the colocalization of G3BP1-ΔRGG, N protein and dsRNA/nsp3 in SARS-CoV-2 WT-infected cells revealed that the recruitment of G3BP1 to RTC is independent of SGs formation pathway (Supplementary Fig. 5c, d).

### G3BP1 recruits 40S ribosomal subunit to viral factories

Viruses rely on the host cell translational apparatus for efficient synthesis of viral proteins and employ a diverse array of mechanisms to gain access to ribosomes for preferential translation of viral mRNAs[42]. Given that G3BP associates with the 40S ribosomal subunit[3,10,43], we proposed that G3BP1 might facilitate viral protein translation. To test this, we used ribopuromycylation staining to map active translation sites in infected cell panels[10,24,44]. As expected, puromycin (PMY) signals were intense and uniformly distributed in all mock infected cells, indicating robust protein synthesis pervading the entire cytoplasm. However, the PMY signal was significantly weaker overall in SARS-CoV-2 WT infected cells, but strongly colocalized with G3BP1 and N protein, revealing that G3BP-N-containing RTC were the locations for the most concentrated translation in infected cells (Fig. 7a, b and Supplementary Fig. 7). Notably, PMY-Max intensity within cells, which reflects translation efficiency at viral factories, was reduced in cells infected with the RATA, resulting in lower viral protein production (Fig. 7d). This was despite similar levels of viral RNA in

RATA and WT infected cells early in infection (Fig. 7c). Additionally, in cells expressing G3BP1-ΔRGG, where the G3BP:40S interaction is absent, PMY signals were not detected despite the strong colocalization of G3BP and N protein (Fig. 7e, f). In summary, these results revealed that G3BP1 plays a role in recruiting the translation machinery to viral factories for the production of nascent viral mRNA at early times post infection.

To achieve a more detailed depiction of this process across different cell lines, we utilized transmission electron microscopy (TEM) to visualize SARS-CoV-2 infection. Consistent with many other positive-strand RNA viruses, SARS-CoV-2 causes restructuring of ER membranes, leading to the formation of DMV, sites for viral RNA synthesis[45,46]. Subsequently, viral mRNA will be exported from DMV to the cytosol for translation[45]. In SARS-CoV-2 WT-infected cells, we observed accumulation of ribosomes around the DMVs (Fig. 7g and Supplementary Fig. 6a). However, in RATA mutant-infected cells, the accumulation was less pronounced, and ribosomes had more diffuse distribution, consequently leading to a decrease in size of the large virus-containing vacuoles and decreased virion production (Supplementary Fig. 6b, c). Moreover, the configuration was influenced by the absence of G3BP, resulting in a dispersed distribution of ribosomes in both WT SARS-CoV-2- and RATA infections in infected cells lacking G3BP (Fig. 7g). These results underscore the significance of G3BP in the regulation of viral mRNA translation by SARS-CoV-2, providing validation for the mechanisms through which G3BP-N complexes facilitate the recruitment of ribosomes to viral factories, and enhance viral protein translation.

### Discussion

G3BP1, a multifunctional protein, can exhibit both proviral and antiviral activities during infection among different viral families[47]. The specific function of G3BP1 is determined by the proteins or RNA with

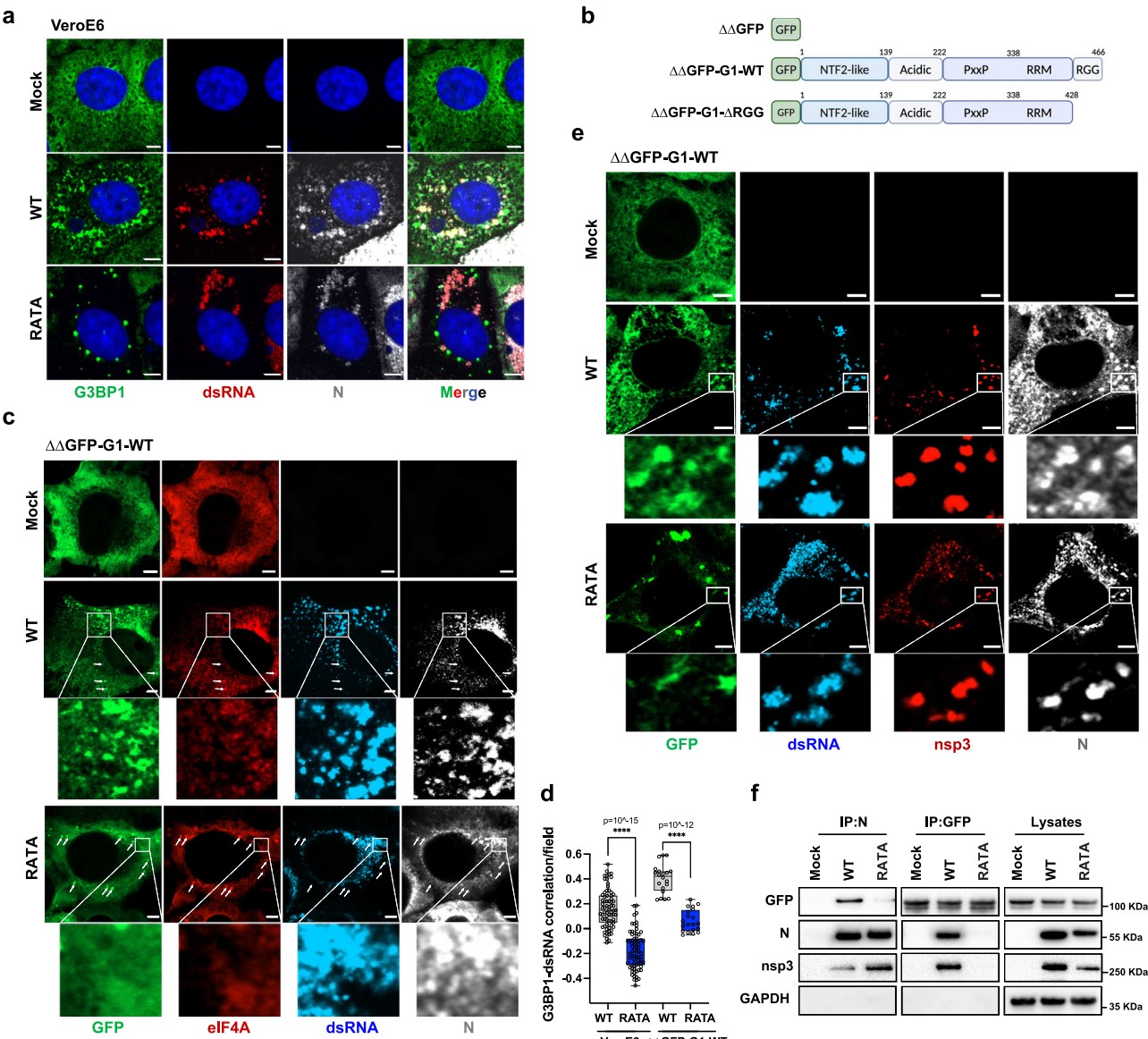

**Fig. 6 | N recruits G3BP1 to RTC early in infection via interaction with pore protein nsp3. a** VeroE6 cells were infected with WT SARS-CoV-2 at 0.5 MOI. Cells were fixed at 6 h and stained for G3BP1 (green), dsRNA (red) and N (gray), Hoechst (blue). Representative images from three independent experiments are shown. Scale bar = 5 μm **b** Schematic of GFP-G3BP1 constructs used for reconstitution of G3BP1/2 double KO cell lines. All lines were subsequently transduced with ACE2 receptor and TMPRSS2. **c** ΔΔGFP-G1-WT cells were infected with WT SARS-CoV-2 WT or RATA at 0.5 MOI for 6 h. Cells were fixed and stained for dsRNA (blue), N (gray) and eIF4A (**c**) or nsp3 (**e**) (red). Representative images from three independent experiments are shown. White arrows indicate SGs. Scale bar = 5 μm. **d** Pearson's correlation coefficients for colocalization of G3BP1 and dsRNA in indicated cells were calculated in CellProfiler (*n* = 71 for VeroE6 cells, *n* = 20 dsRNA-positive fields for ΔΔGFP-G1-WT cells). Box plots bound the interquartile range divided by the median, with the whiskers extending from the minimum to the maximum values. **f** ΔΔGFP-G1-WT cells were infected with WT virus or RATA mutant at 0.01 MOI for 24 h. Cells were lysed and immunoprecipitated with GFP or N antibody for immunoblotting as indicated.

which it interacts. When bound to cGAS for DNA virus, or RIG-I for RNA virus, G3BP1 promotes the IFN-β responses, thereby inhibiting viral replication[48,49]. Additionally, as a nucleating protein of SGs, G3BP1 binds mRNAs, 40S ribosomal subunits, initiation factors (eIF2, eIF3, eIF4), and RNA-binding proteins (e.g., poly(A) binding protein, Caprin1) for SGs assembly and protein translation arrest after viral infection. Our data indicate that SGs were induced via the PKR/PERK-eIF2α signaling pathway, which corresponded with reduced total cellular protein translation during early SARS-CoV-2 infection. However, SARS-CoV-2 N protein has evolved countermeasures to neutralize the antiviral functions of G3BP1 by targeting it directly to disassemble SGs[14–16,23]. The mechanism involves the high binding affinity between G3BP1 and the N protein, which can sequester G3BP1 and inhibit its

interaction with other key SG proteins. This hypothesis is supported by our lysate-based LLPS experiment and a peptide competition assay[17]. Additionally, we observed that the N protein hijacks G3BP1 with viral dsRNA at sites of active viral replication at early infection, where no other SG proteins, such as eIF4A, accumulate, suggesting the formation of proviral assemblies involving G3BP1.

N protein is essential for viral replication and assembly and is highly abundant in SARS-CoV-2-infected cells[50,51]. It is recruited at the DMV pores by nsp3[41], and there forms phase separated droplets with RNA-dependent RNA polymerase (RdRp) complex of nsp12, nsp7, and nsp8 and RNA, potentially aiding in the entry of viral genomic RNA into the DMV, initiating the initiation and/or elongation of viral RNA synthesis[33]. A recent report shows that nsp3 also binds fragile X

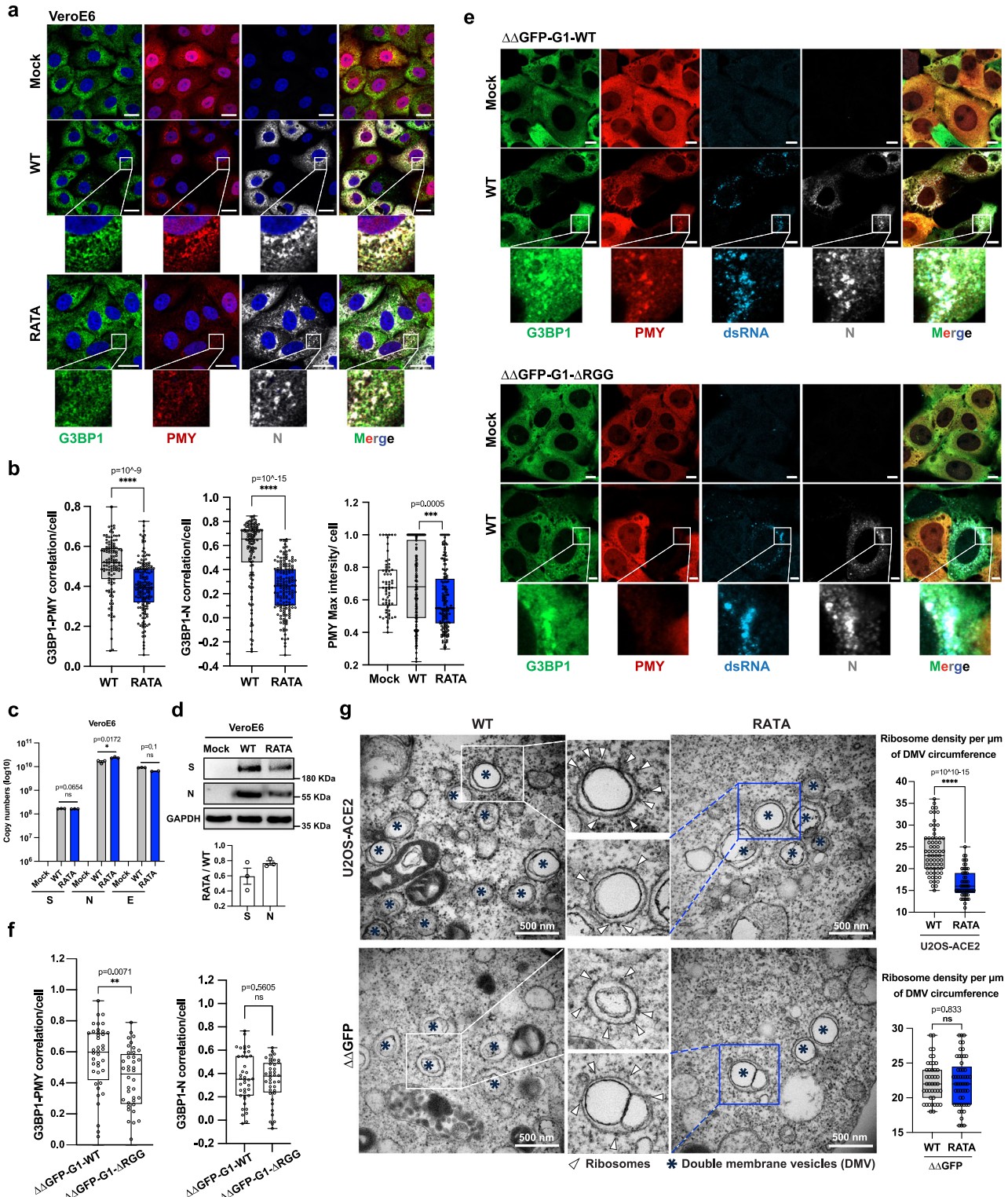

a VeroE6

b

c VeroE6
d VeroE6

e ΔΔGFP-G1-WT

ΔΔGFP-G1-ΔRGG

f

g

◁ Ribosomes   ✳ Double membrane vesicles (DMV)

mental retardation proteins[52], likely also contributing to the disassembly of SGs and phase transition of proteins concentrated at the DMV periphery. Our data demonstrate that G3BP is recruited to viral factories by N protein, where it forms a complex with the N protein, viral dsRNA, and nsp3. Subsequently, we observed high levels of localized translation at the viral factories due to the interaction at G3BP1 RGG domain with cellular 40S ribosomal subunits[3]. Probably the translation pre-initiation complexes that accumulate in infection-induced SGs at very early times, are then recruited, along with G3BP to the viral factories and are made available for translation of nascent viral mRNAs produced at those sites. Interestingly, other positive strand RNA viruses, from the alpha- and noroviruses have also evolved similar strategies to manipulate G3BP for efficient viral mRNA translation[8,10] suggesting it may be a common mechanism to take over the translation machinery for efficient production of viral proteins. Thus, the ΦxFG motif binding groove on the NTF2L is a promising target for broadly acting antiviral interventions. Notably, the proviral roles of G3BP in chikungunya virus and in norovirus infections are critical and viral replication is severely compromised or undetectable in its absence[8,10], while our present work shows that

**Fig. 7 | G3BP1 recruits 40S ribosomal subunit to viral factories. a** VeroE6 cells were infected with WT SARS-CoV-2 or RATA mutant at 0.5 MOI for 6 h. Cells were incubated with PMY (20 μg/mL) for 2 min before fixation and stained for G3BP1 (green), PMY (red), N (gray), Hoechst (blue). Representative images from three independent experiments are shown. Scale bar = 20 μm. **b** Correlations of G3BP1-PMY, or G3BP1-N, and PMY max intensity were calculated in CellProfiler based on Pearson's correlation coefficient for infected cells ($n = 126$ in WT, $n = 160$ in RATA, $n = 66$ in Mock). Box plots bound the interquartile range divided by the median, with the whiskers extending from the minimum to the maximum values **c** VeroE6 cells were infected with SARS-CoV-2 WT or RATA mutant at 0.05 MOI for 6 h, cells were collected for RT-qPCR to quantify viral RNA expression ($n = 3$ biological replicates) and **d** for immunoblotting with indicated antibodies. Representative images from three independent experiments are shown. Quantification of western blot was performed using Image J. Bar chart in (**c**, **d**) are presented as mean values ± SEM as appropriate. **e** Indicated cell lines were infected with SARS-CoV-2

WT at 0.5 MOI for 6 h. Cells were incubated with PMY (20 μg/mL) for 2 min before fixation and stained for PMY (red), N (gray), dsRNA (blue). Representative images from three independent experiments are shown. Scale bar = 5 μm. **f** Correlations of GFP-G3BP1-PMY, or GFP-G3BP1-N were calculated in CellProfiler based on Pearson's correlation coefficient for $n = 40$ infected cells. Box plots bound the interquartile range divided by the median, with the whiskers extending from the minimum to the maximum values **g** Indicated cells were infected with SARS-CoV-2 WT or RATA mutant at 0.5 MOI for 10 h and processed for transmission electron microscopy. DMV are indicated with asterisks and DMV-associated ribosomes by white triangles. Scale bar = 500 nm. Quantification of ribosome density per DMV (the number of ribosomes attached to DMV divided by length of DMV perimeter) was calculated in Image J. Each dot represents a DMV (U2OS WT = 66, U2OS RATA = 57, ΔΔ WT = 43, ΔΔ RATA = 53). Box plots bound the interquartile range divided by the median, with the whiskers extending from the minimum to the maximum values.

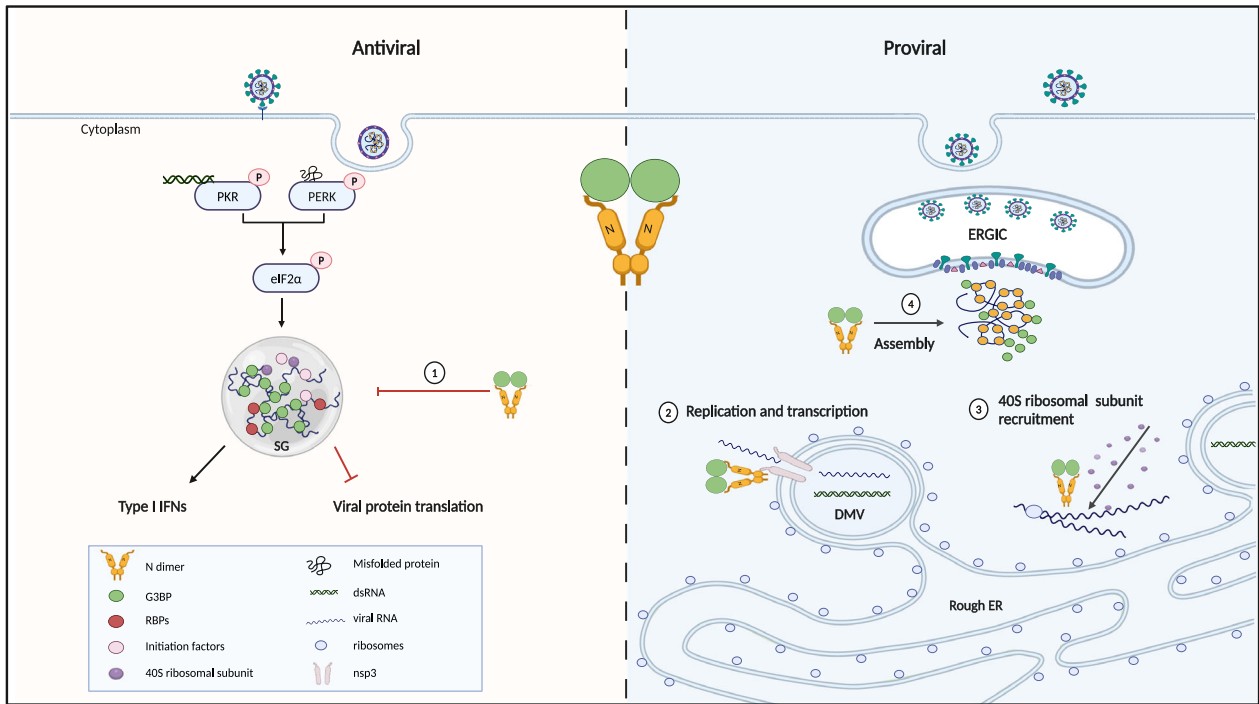

**Fig. 8 | The antiviral and proviral roles of G3BP in SARS-CoV-2 infected cells.** SARS-CoV-2 infection activated PKR/PERK-eIF2α serves as a protective mechanism in host cells, leading to the G3BP dependent SGs assembly. However, SARS-CoV-2 N protein hijacks G3BP, contributing to the enhancement of SARS-CoV-2 replication across multiple stages of the replication cycle: (**1**) G3BP-N interaction mediates the disassembly of SGs. (**2**) Early in infection, the N protein recruits G3BP to nsp3 at the RTC, potentially aiding in viral RNA synthesis and transcription; (**3**) The G3BP-N complex recruits 40S ribosomal subunits to viral factories for efficient viral protein translation; (**4**) G3BP promote the LLPS of N, facilitating SARS-CoV-2 virus assembly. Created in BioRender.

the proviral role in SARS-CoV-2 infection is accessory but not critical. Moreover, N protein also facilitates the compaction of N protein and its large 29.9 kb viral genomic RNA into a highly structured viral RNA-protein complex, facilitating its packaging into virions through subsequent N-M protein interactions[36]. Our data show that G3BP1 facilitates the LLPS of N with RNA in a dose-dependent manner. We propose that the facilitation of LLPS of N by G3BP serves a dual role. Firstly, it aids in the recruitment of RNA and host proteins for efficient RNA replication and transcription at the RTC. Secondly, it concentrates viral genomic RNA and structural proteins, promoting the assembly of virus particles. This is supported by a recent report highlighting the G3BP's role in virus assembly and its integration into virus particles[53].

This study presents a comprehensive examination of the importance of G3BP for SARS-CoV-2 infection both in vitro and in vivo (Fig. 8). The works contributes to an emerging view of G3BP as both

antiviral (as a component of SGs) and, when sequestered to replication complexes, as a proviral factor involved in viral gene expression and replication. The loss of G3BP binding site on SARS-CoV-2 N protein attenuates viral replication in different cell lines and pathogenesis in mouse model of severe COVID-19, supporting the concept that G3BP-N interaction is crucial both for evasion of host defense and for efficient viral gene replication and expression. However, host defense and viral replication are complex and intricate processes. Given the diverse and multifaceted properties of G3BP1, we propose that antiviral SGs and proviral granules involving G3BP1 may coexist and potentially both affect viral replication and gene expression. This might explain why knockdown or knockout of G3BP were in some studies shown to boost SARS-CoV-2 infection[21], while in other cases, the opposite or no effect was observed[40,53]. However, in RATA-infected cells, G3BP predominantly contributes to the formation of consistent SGs, thus resulting significant attenuation both in vitro and in vivo. We also

showed that RATA has the capacity to sufficiently activate immune responses, subsequently offering protective immunity upon rechallenge. This diminished pathogenicity and robust immune activation suggests that mutation of G3BP-interaction domains could be viable strategies for live attenuated vaccine development across multiple virus families.

## Methods

### Cell lines

VeroE6 cells (ATCC-CRL-1586), MA104 (ATCC-CRL-2378.) and HEK-293T cells (ATCC-CRL-1573) were maintained in Dulbecco's modified Eagle medium (DMEM) (Sigma, #D6429) supplemented with 10% fetal bovine serum (FBS) (Sigma, #F9665), 1% penicillin–streptomycin (Sigma, #P4333) and 1% L-glutamine (Sigma, #G7513). Baby hamster kidney (BHK-21) cells (ATCC CCL-10) were maintained in Glasgow's modified Eagle's medium (GMEM) (Sigma, #G5154) supplemented with 10% FBS, 10% tryptose phosphate broth (Sigma, #T8159), 20 mM HEPES (Sigma, #H0887), 1mM L-glutamine and 1% penicillin–streptomycin. Cells were cultured at 37 °C in a humidified incubator with 5% $CO_2$.

U2OS-ACE2 cells were generated by transduction of human osteosarcoma (U2OS) cells (ATCC HTB-96) with lentivirus expressing ACE2-TMPRSS2 and selection with PMY (Thermo Fisher Scientific, #A1113803). ΔΔGFP, ΔΔGFP-G1-WT, and ΔΔGFP-G1-ΔRGG cells were generated by transduction of corresponding cell lines (described in ref. [10]) with lentivirus expressing ACE2-TMPRSS2 and selection with PMY. Cells were then sorted using fluorescence-activated cell sorting (FACS) on a BD FACSAria Fusion system. The sorting was based on the expression of enhanced green fluorescent protein (EGFP) and RBD-AS635P binding specific to ACE2[54] (Supplementary Fig. 8). Cells were maintained in DMEM supplemented with 10% FBS, 1% penicillin–streptomycin and 1% L-glutamine.

Lentivirus was produced by transfection of HEK-293T cells with pWPI-IRES-Puro-Ak-ACE2-TMPRSS2, VSV-G and Gag-Pol using Lipofectamine 2000 Reagent (Thermo Fisher Scientific, #11668019). pWPI-IRES-Puro-Ak-ACE2-TMPRSS2 was a gift from Sonja Best (Addgene plasmid #154987; http://n2t.net/addgene:154987; RRID:Addgene 154987).

### Generation of RATA mutant virus

Both WT virus and RATA mutant were derived from SARS-CoV-2 Wuhan-Hu1 (MT926410) infectious clone[30]. For RATA construction, the mutation was introduced into a subclone pCC1-4K-FR1 by using PCR with primers RATA-F and RATA-R (Supplementary Table 1). Thereafter, plasmids containing WT or RATA mutant fragments were digested by restriction enzyme, and then were purified and ligated in vitro to assemble the full-length cDNA using Gibson Assembly master mix (New England Biolabs, #E2611). The plasmid was verified by restriction enzyme digestion and Sanger sequencing (Eurofins Scientific). To rescue the infectious virus, BHK-21 cells were transfected with plasmid DNA (containing icDNA of SARS-CoV-2) using Lipofectamine LTX (Thermo Fisher Scientific, #15338100). After 3 days, supernatant was transferred to VeroE6 cells to propagate viruses. At 3 days post infection, P0 stock virus were harvested for sequencing using primers listed in Supplementary Table 1.

### Plaque assay

Monolayers of cells were infected with serial 10-fold dilutions of virus suspension for 1 h, and then washed with PBS (Sigma, #D8662) before addition of 1 mL of prewarmed overlay (2% methylcellulose (Sigma, #C5013): propagation media containing 2% FBS = 2:3). At 72 hpi, cells were fixed with 4% formaldehyde (Sigma, #F8775) and stained with crystal violet solution (HistoLab, CL.42555) after removal of the overlay. Plaques were manually counted. The determination of all virus titers was performed in triplicate.

### Infection

Virus titrations for infection experiment were determined by plaque assay. As for in vitro infection, cells at 80% of confluency were infected with virus in infection media supplemented with 0.2% BSA, 2 mM L-glutamine and 20 mM HEPES for 1 h at 37 °C, 5% $CO_2$. The infection media were then removed and washed with warmed PBS before adding warmed propagation media containing 2% FBS.

K18-hACE2 transgenic mice were purchased from the Jackson Laboratory and maintained as a hemizygous line. All mice were 18–20 weeks old at the start of the study, and experiments were conducted in BSL3 facilities at the Comparative Medicine department (KM-F) at Karolinska Institutet. Ethical permits for studies of virus infection were obtained from the Swedish Board of Agriculture (10513-2020). The mice were maintained in a specific pathogen-free facility with controlled conditions, including a stable temperature of 25 °C and humidity levels between 40% and 60%, and a 12-h light/dark cycle. Mice were housed in individually ventilated cages with access to food and water ad libitum. Enrichment materials, such as shredded cardboard and paper rolls, were provided to enhance their living environment. Health monitoring was carried out daily by trained personnel, with weekly cage and water changes ensuring proper hygiene. Mice were challenged intranasally with 100 PFU or 1000 PFU SARS-CoV-2 in 40 µL of PBS following isoflurane sedation. During the experiment, weight loss, changes in general health, breathing, body movement, and posture were monitored. Mice were euthanized when they reached 20% weight loss or when movement was greatly impaired and/or they experienced difficulty breathing that was considered to reach a severity level of 0.5 on Karolinska Institutet's veterinary plan for monitoring animal health.

### Histology and Immunohistochemistry

Mice lungs were fixed with 4% formaldehyde in PBS overnight at 4 °C and transferred to 70% ethanol (VWR, #20824.296) at 4 °C for storage. Samples were then dehydrated in a graded series of ethanol from 70% to 99%. The samples were cleared with clearing agent (Histolab, #14250), embedded in melted hot paraffin, and hardened on ice. Thin sections of 5 µm thickness were sliced using a rotary microtome outfitted with disposable steel knives, flattened on a heated water bath, transferred onto microscope slides, and dried. Sections were deparaffinized by incubating in xylene (Histolab, #02070) and rehydrated by passing through a series of descending ethanol concentrations (100%, 95%, 70%). Sections were stained with hematoxylin, which colors cell nuclei blue, and eosin, which imparts a pink hue to cytoplasm and other structures. After staining, sections were washed in water, dehydrated, and cleared to prepare them for mounting with a coverslip. Images were taken by a Ziess Axioplan light microscope with an Olympus SC30 camera.

For immunohistochemistry staining, paraffin-embedded tissues were de-paraffinized by xylene and rehydrated with sequential incubation with 99%, 95%, 70% of ethanol, followed by blocking with 5% bovine serum albumin (BSA) (Sigma, #A4503). Sections were then incubated with primary antibody against SARS-CoV-2 N protein (Abcam, ab271180, 1:3000) overnight at 4 °C, washed, and incubated with appropriate secondary antibodies Alexa Fluor 488 (Thermo Fisher Scientific, A21206, 1:1000). After washing with PBS, sections are dehydrated and cleared to prepare them for mounting with a coverslip. Images were taken by Zeiss LSM800-airy confocal microscope equipped with a pulsed white light laser and a Zeiss DIC Prism III PA 63x/1.40 oil objective. Images were processed by Zeiss ZEN microscope software.

### Immunofluorescence assay

Cells seeded on cover glasses (VWR, #631-1554) were washed with PBS three times and fixed with 4% formaldehyde in PBS for 20 min at room temperature (RT), permeabilized in methanol (Sigma, #179957)

for 10 min at −20 °C and blocked with 5% BSA in PBS at RT for 1 h. Primary antibody and secondary antibody were diluted in blocking solution. Cells were incubated with primary antibody at 4 °C overnight, followed by 1 h incubation with secondary antibodies (Supplementary Table 2). Cover glasses were mounted on glass slides by mounting media and imaged by Zeiss LSM800-airy confocal microscope equipped with a pulsed white light laser and a Zeiss DIC Prism III PA 63x/1.40 oil objective. Images were processed by Zeiss ZEN microscope software.

### Immunoprecipitation

Cells were washed with ice cold PBS and lysed in lysis buffer (Thermo Fisher Scientific, #87787) supplemented with protease and phosphatase inhibitors (Thermo Fisher Scientific, #1861279, #78427) on ice for 10 min. Lysates were collected and cleared by centrifuging at 13,000 × g for 10 min at 4 °C, and incubated with GPF-trap beads (ChromoTek, #gta) for 1 h with rotating at 4 °C, or incubated with antibody (Supplementary Table 2) for 1 h and Protein A Magnetic Beads (Cytiva, #28951378) overnight at 4 °C under rotation. Beads were then washed three times with cold lysis buffer and eluted in 2x NuPAGE LDS sample buffer (Thermo Fisher Scientific, #NP0007) containing 50 mM DTT (Sigma, #D0632), heated at 95 °C for 10 min and analyzed by SDS-PAGE and western blotting.

### SDS-PAGE and western blot

Samples were collected and denatured in LDS sample buffer (Thermo Fisher Scientific, #NP0007) for sodium dodecyl sulfate polyacrylamide gel electrophoresis (SDS-PAGE) on NuPAGE 4–12% Bis-Tris polyacrylamide gels (Thermo Fisher Scientific, #NP0321BOX) and transferred onto 0.45 μm Immun-Blot PVDF membrane (Bio-Rad, #1704272). Membranes were blocked with 5% of skim milk powder (Sigma, #70166) in Tris-buffered saline with 0.05% Tween 20 (TBST) and incubated with primary antibodies at 4 °C overnight, and horseradish peroxidase (HRP)-coupled secondary antibodies for 1 h at RT (Supplementary Table 2). Chemiluminescence (Thermo Fisher Scientific, #34580) was applied to the surface of PVDF membrane and Image lab was used for image detection and procession. All SDS-PAGE data are representative of at least three independent experiments. Quantification of western blot was analyzed by Image J.

### Quantitative reverse transcription polymerase chain reaction (RT-qPCR)

Cells were homogenized in TRIzol reagent (Thermo Fisher Scientific, #15596018), and RNA is extracted through phase separation with chloroform (Thermo Fisher Scientific, #C/4960/PB08), followed by RNA precipitation with isopropanol (Sigma, #I9516), and washing with ethanol. The RNA pellet is then resuspended in DEPC water (Thermo Fisher Scientific, #R0601). Subsequently, cDNA is synthesized from the purified RNA using SuperScript™ IV reverse transcription kit (Thermo Fisher Scientific, #18091050). The qPCR reaction mix is prepared with SYBR Green PCR Master Mix (Thermo Fisher Scientific, #A46109), specific primers (Supplementary Table 1), cDNA template, and amplification is performed under optimized cycling conditions on CFX384 Touch Real-Time PCR System. Data were analyzed by Bio-Rad CFX Manager software.

### Ribopuromycylation assay

Ribopuromycylation assay[10,24,44] was performed by infecting cells with indicated virus at an MOI of 0.5 for 6 h and treatment with 20 μg/mL of PMY for 2 min at 37 °C, 5% CO$_2$. Following incubation, cells were rinsed with PBS and fixed with 4% formaldehyde in PBS for 20 min at RT, permeabilized in methanol for 10 min at −20 °C and blocked with 5% BSA in PBS at RT for 1 h. After blocking, cells were incubated with antibodies (Supplementary Table 2). Images were taken by Zeiss LSM800-airy confocal microscope and processed by ZEN microscope software.

### Recombinant protein purification

DNA fragments encoding for human G3BP1, the SARS-CoV-2 N protein, and the SARS-CoV-2 N (RATA) mutated protein were inserted into the pET30 expression vector using ligation-independent cloning[55]. A TEV cleavage site was introduced between the N-terminal his 6 tag and each fusion protein. All constructs were validated by sequencing using primers (Supplementary Table 1) (Eurofins Scientific). G3BP1 full-length protein was expressed and purified from *E. coli* BL21 cells (Sigma, #69450) and purified under native conditions. *E. coli* were grown to OD 600 of 0.8 and induced with 0.5 mM IPTG at 22 °C overnight. Pelleted cells were resuspended in lysis buffer (50 mM HEPES pH 7.5, 300 mM NaCl, 1 mM DTT, protease inhibitor (Roche, #11873580001) and Benzonase (Sigma, #E8263)). Following sonication, lysates were pelleted at 40,000 g at 4 °C for 30 min. His6-TEV-G3BP1 was affinity-purified by Ni-NTA agarose (Qiagen, #30210) and the eluted proteins were incubated with TEV protease (PSF, KI, Stockholm) at 4 °C overnight. Fully cleaved proteins were further purified on a HiTrap Heparin column (Cytiva, #17040601). Collected fractions were analyzed by coomassie staining and SDS-PAGE, and thereafter pooled and concentrated. All proteins were further purified on a Superdex 200 16/200 column (Cytiva, #28989335), equilibrated in SEC buffer (50 m HEPES, pH 7.5, 300 mM NaCl, 1 mM DTT). Fractions were analyzed by SDS-PAGE, pooled, concentrated, filtered, flash frozen in liquid nitrogen, and stored at −80 °C. The SARS-CoV-2 N and SARS-CoV-2 N RATA proteins were purified following similar protocols. Each target protein was affinity-purified by Ni-NTA agarose and further isolated on a Superdex 200 16/200 column. Fractions were analyzed by coomassie staining and SDS-PAGE, concentrated, and stored at −80 °C.

### Lysate based liquid−liquid phase separation (LLPS) assay

$1 \times 10^7$ cells of the indicated lines were lysed for 5 min at RT in 800 μL lysis buffer (50 mM HEPES, 0.5% NP40, protease inhibitor, and 2.5% murine RNase inhibitor (New England Biolabs, #M0314)) and centrifuged for 5 min at 20 000 g at 20 °C. Separately, recombinant proteins were mixed with protein dilution buffer (50 mM HEPES pH 7.4, 400 mM NaCl, 1 mM DTT) in 8 μL volume. Where indicated, 8 μL of primary and secondary antibodies were diluted in the lysis buffer and incubated for 30 min at RT. Total protein concentration was adjusted to 6.0 mg/mL, in the collected cell-lysate supernatant. Then 40 μL of cell lysate was added to the samples and 50 μL of the mixture was immediately transferred to an 18-well microscope chamber slides (IBIDI, #81816) and incubated for 1 h at RT. Images were taken with a Zeiss LSM700 Confocal Microscope equipped with a pulsed white light laser and a Zeiss Plan-Apochromat 63X/1.40 oil DIC objective. Condensates were analyzed with software Fiji.

### Transmission electron microscopy (TEM)

Cells were immersion fixed in 2.5% glutaraldehyde (Sigma, #G5882) at RT for 1 h and stored at 4 °C or were rinsed in PBS followed by post fixation in 2% osmium tetroxide (Sigma, #419494) at 4 °C for 2 h. After stepwise dehydration in ethanol and acetone (Sigma, #179124), the samples were resin infiltrated and finally embedded in LX-112 (Ladd Research, #21210). Ultrathin sections (~80–100 nm) were prepared using an EM UC7 (Leica) placed on formvar slot grids and contrasted with uranyl acetate followed by lead citrate. The grids were examined in a HT7700 transmission electron microscope (Hitachi High-Technologies,) at 80 kV and digital images were acquired using a 2k × 2k Veleta CCD camera (Olympus Soft Imaging Solutions).

### Statistical analysis and reproducibility

The statistical details for all experiments, including error bars, statistical significance, and precise n numbers, are provided in the figure legends. Statistical comparisons between two groups were conducted using the unpaired *t*-test (two tailed). For multiple group comparisons,

**Article** https://doi.org/10.1038/s41467-024-54996-3

the one-way ANOVA test (two-sided) was employed to compare the mean of each column with the mean of a control column. Statistical analysis was conducted using GraphPad Prism 10 software. ns, $P > 0.05$; *, $P \leq 0.05$; **, $P \leq 0.01$; ***, $P \leq 0.001$, ****, $P \leq 0.0001$.

### Reporting summary

Further information on research design is available in the Nature Portfolio Reporting Summary linked to this article.

## Data availability

The study involves no large datasets that should be uploaded to any depository. Raw data from our experiments can be found in the Source Data file. 7SUO [https://www.rcsb.org/structure/7SUO] Source data are provided with this paper.

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

## Acknowledgements

We thank Lars Haag at the KI electron microscopy unit (EMil) for help with transmission electron microscopy and Changil Kim and Ben Murrell for help with setting up in vivo experiments. All work with infectious virus was performed in Biosafety Level 3 laboratory at Karolinska Institutet Core facility. Work in the G.M.M. laboratory is supported by project grants from the Swedish Research Council (grant nos. 2020-04706 and 2022-03045), the Swedish Cancer Society (grant no 21 1779 Pj), and Sygeforsikring Danmark. S.L. was supported by a Chinese Scholarship Council fellowship. M.G. was supported by a grant from the Swedish Foundation for Strategic Research (#UKR22-0064).

## Author contributions

Conceptualization: S.L., G.M.M. Methodology and Investigation: S.L., M.G., L.P.V., X.H., R.S., M.W., M.D.P., E.U., and A.M. Supervision: M.D.P., J.M.C., A.M., A.A., and G.M.M. Funding acquisition. G.M.M. Writing—original draft: S.L. and G.M.M. Writing—review and editing: S.L., A.M., A.A., and G.M.M.

## Funding

## Competing interests
The authors declare no competing interests.
