## [Transparent Peer Review file · Nature Communications]

SARS-CoV-2 N protein recruits G3BP to double membrane vesicles to promote translation of viral mRNAs

Corresponding Author: Professor Gerald McInerney

Version 0:

Reviewer comments:

Reviewer #1

(Remarks to the Author)
Summary/Key Results

The manuscript Long et al. investigates the pro-viral role of G3BP1/2 during SARS-CoV-2 infection. Previous studies have established that the SARS-CoV-2 Nucleocapsid (N) protein binds with G3BP1/2, inhibiting stress granule formation (which is believed to be antiviral). However, whether G3BP1/2 is itself pro- or anti-viral is a matter of some debate. To investigate the importance of N:G3BP1/2 binding, the authors use reverse genetics to generate a recombinant virus with a two amino acid substitution in the N protein (N: I15A; F17A). Compared to infection with wild type SARS-CoV-2, the authors mutant exhibits reduced replication, N:G3BP1/2 binding, and increased stress granule formation.

The most interesting findings of this article are mechanistic. The authors clearly demonstrate that G3BP1 co-localizes with NSP3 on double membrane vesicles and that this interaction requires N:G3BP1 binding. Furthermore, they provide evidence that this results in enhanced translation at RTCs, evidenced by the co-localization of 40S ribosomal subunits, ribopuromycylation experiments, and the number of ribosomes localized to DMVs by TEM.

Overall, this manuscript is both interesting and scientifically sound. In particular, the last 1/3 (Fig. 5 and 6) of the paper linking N:G3BP1 binding with enhanced translation at DMVs is novel. Pro-viral functions of G3BP1 during SARS-CoV-2 infection have been proposed, but the data presented here are the first to elucidate a concrete mechanism.

Despite this article's strengths, there are some major weaknesses. First, the overall novelty of the paper is lessened by the fact that a similar article, Yang et al. 2024, has already been published. In Yang et al., the authors produced a single substitution mutant (F17A) SARS-CoV-2, and described the effects on in vitro replication, G3BP1/2 binding, phase separation, stress granule formation, and in vivo infection. Given the similarity in the mutants (F17A vs I15A/F17A) and that the results and conclusions are almost identical (they differ only in magnitude), most of the data presented in Figures 2 – 4 are merely a confirmation of prior work. At minimum, the authors need to cite this past work.

In conclusion, the mechanistic work present in the figures 5 and 6 of this manuscript are novel, interesting, and important. However, this novelty is harmed by the fact that the data in Figures 2 – 4 is largely a confirmation of a manuscript of another group, which the authors do not cite. The authors can improve their manuscript by a) emphasizing and expanding upon those findings which are novel and b) citing Yang et al. and addressing the minor points of difference between the two studies.

Specific Critiques:

Major:

1. Novelty: Yang et al. have already published on a recombinant SARS-CoV-2 mutant that disrupts N:G3BP1/2 binding. Their mutant contains 1 of the 2 mutations used by the authors (F17A) and show similar results. Thus, Yang et al. has essentially already described the following data included in this manuscript:

a. structure of the N:G3BP interface within the NFT2L domain (Fig. 2A)

- b. disruption of N:G3BP1 binding through the F17A mutation (Fig. 2C)
- c. increased stress granules in cells infected with the F17A mutant compared to WT (Fig. 2D)
- d. decreased viral replication in vitro in the F17A mutation (Fig. 2G)
- e. decreased pathogenesis and replication in vivo using a small animal model (Fig. 3B, 3C)
- f. That G3BP1 facilitates LLPS, which is disrupted by the F17A mutation (Fig. 4)

While the mutants are not identical (F17A alone in Yang, vs I15A and F17A in the present manuscript) the same conclusions are reached. At minimum, Yang et. al. should be cited. Overall, this reduces the novelty of the research significantly.

2. Specificity of Mutant: While the underlying conclusions of Yang et. al are concordant with the authors work, there is a major difference in the magnitude of the effects seen. For example, Yang et. al. reports ~1-log reduction in virus replication with their mutant (F17A) but the present study reports ~2-log reduction with theirs (I15A, F17A). The Yang et al. manuscript spends a great deal of time demonstrating that the F17A mutation is highly specific, disrupting SARS2 N:G3BP1/2 binding while not affecting SARS2 N's affinity to any other cellular proteins according to mass spectrometry analysis. Thus, one possible explanation for the differences in the magnitude of phenotypes between the two studies is that the addition of I15A results in off target effects. Control experiments addressing the specificity of the double mutant are needed to alleviate these concerns.

3. Viral replication in K18-hACE2 mice. Fig. 3 compares the pathogenesis of SARS2 WT vs the I15A, F17A mutant, demonstrating that the I15A, F17A mutation reduces weight loss, H&E staining, and antigen staining relative to WT. The absence of viral lung titer is a striking omission. Viral lung titers between SARS2 WT and the I15A, F17A mutation should be examined to confirm replication differences in vivo.

4. Translation of viral proteins: As mentioned in the summary, the authors finding that N re-localizes G3BP1 from stress granules to replication transcription complexes to enhance translation is a major strength and a real step forward for the field. However, when digging into the details of their model, the authors propose that the ribopuromycylation staining co-localizing with NSP3 indicates enhanced translation of viral genes as they emerge from DMVs. While plausible, this is not adequately supported. Looking at Fig. 2C, the protein levels of N itself are not affected much by the I15A, F17A mutation. This should be explained, or the levels of other viral proteins should be examined. Given the normal abundance of N within infected cells, viral factors translated from longer subgenomic transcripts may be affected more drastically and validate their model.

Minor:

1. While the microscopy data presented in Fig.5 and Fig 6 (IF and TEM) is convincing, it would be improved by any sort of quantification that the authors would be able to provide. For instance, the number of ribosomes per DMV in Fig. 6C, etc.
2. The scale bars on all the IF images throughout the manuscript are too small to read.

Citations

Yang Z, Johnson BA, Meliopoulos VA, Ju X, Zhang P, Hughes MP, Wu J, Koreski KP, Clary JE, Chang TC, Wu G, Hixon J, Duffner J, Wong K, Lemieux R, Lokugamage KG, Alvarado RE, Crocquet-Valdes PA, Walker DH, Plante KS, Plante JA, Weaver SC, Kim HJ, Meyers R, Schultz-Cherry S, Ding Q, Menachery VD, Taylor JP. Interaction between host G3BP and viral nucleocapsid protein regulates SARS-CoV-2 replication and pathogenicity. *Cell Rep.* 2024 Mar 26;43(3):113965. doi: 10.1016/j.celrep.2024.113965. Epub 2024 Mar 15. PMID: 38492217.

Reviewer #2

(Remarks to the Author)

This manuscript addresses the mechanisms by which G3BP proteins affect the ability of SARS2 to infect human cells. The work concludes that the G3BP proteins interact with the SARS2 N protein to promote the translation of viral RNAs that are emerging from the membrane bound replication organelles. Although this is potentially an interesting manuscript, as detailed below, significant additional analyses would be required to support the major conclusions.

This review is from Roy Parker and I would be happy to clarify these comments for the authors if needed.

Specific Comments:

- 1) The novel conclusion of this work is that a complex of G3BP-N is targeted to the pore complex of double membrane replication organelles to recruit ribosomes to the emerging viral mRNAs. I am not yet convinced of this conclusion given the current data and make the following comments/suggestions.
 - a) The first argument for this function is that viruses with an N protein mutation that blocking interact with G3BP proteins showed reduced replication (similar to what was observed in Yang et al., 2024, Cell Reports). However, it remains possible that G3BP inhibits SARS replication and in the absence of N proteins inhibiting G3BP, there is an inhibitory effect on SARS (as suggested at least in part in Burke et al., 2024, Science Advances). I realize the field is split on this issue, but the authors could clarify this issue by examining WT and RATA SARS replication in WT and *g3bpΔΔ* cells. This would clarify whether G3BP is a host factor, or just limits SARS infection, and how those putative roles are affected by interaction with the N protein. Without clearly demonstrating G3BP is a required host factor it is difficult to argue the N-G3BP protein interaction is required for viral growth.
 - b) The second argument for this function is that G3BP overlaps with dsRNA and the N protein earlier in infection. This

experiment could be improved by i) showing the individual channels (at least in supplement) since it is not possible to assess how significant these overlaps are with just one small zoom showing individual channels, and ii) some type of quantification at all the time points. I find it notable that at 12 hours it looks like G3BP is excluded from the area of dsRNA, which is similar to what we observed (Burke et al., 2024, Science Advances). How would exclusion of G3BP from replication areas at this time point fit with the proposed model?

c) The third argument for this function of GFP-G3BP is the overlaps of dsRNA, N, and the Nsp3 protein (as a marker of replication organelles). I make two comments on this experiment. i) I am concerned this staining pattern could be affected by the GFP tag on G3BP1. I suggest this possibility because we observed a different distribution of GFP-G3BP and untagged G3BP in SAR infected cells (see Figure below). Because of this difference, we avoided the use of GFP tagged G3BP for SARS experiments. At a minimum, the authors need to show this subcellular distribution of proteins is not affected by the GFP tag. ii) In addition, however this experiment is performed (e.g. GFP or IF), the experiment could be improved by showing the individual channels (at least in supplement) and quantifying the extent of co-localization.

d) The fourth argument for G3BP promoting local translation is that puromycin labeling overlaps with G3BP IF. It was my understanding that puromycin labeling can no longer be relied upon to identify sites of local translation due to rapid diffusion of released peptides, even when using translation elongation inhibitors in conjunction (Enam et al., 2020, eLife). Given this caveat, additional observations would be needed to make a robust conclusion for G3BP marking the sites of translation. (Can you see ribosome clusters overlapping with G3BP by CLEM or EM with gold labeled antibodies?)

e) The final argument for this function of G3BP is EM imaging showing differences between WT and RATA SARS infection. The key observations are that in RATA mutants and *g3bpΔΔ* cell lines the ribosomes are more diffuse and less concentrated around the double membrane viral organelles. While a single EM image can be interesting, to make these points robustly, those differences need to be quantified in some manner (beyond just the size of the LVCVs).

Additional Issues:

2) How do the authors identify the infected cells in Figure 1a/b since the N protein IF is dim at 6 hours. Is the cell making stress granules at 6 hours infected? This should be clarified.

3) The specific model the authors put forth predicts that in the RATA virus (or in *G3BPΔΔ* cell lines), the translation efficiency of the viral RNAs would be low early in infection. This could be directly tested by measuring the rate of viral protein production and the levels of viral RNAs at the same time point and comparing WT to RATA.

Figure for authors showing differences in endogenous G3BP1 distribution and the distribution of GFP-G3BP1:

(Since the website will not let me insert the Figure, I will email it directly to the authors with a copy of my review.

Note: I also uploaded to the review site as well.)

Reviewer #3

(Remarks to the Author)

In this article, Long and colleagues generate a SARS-CoV-2 mutant (RATA) that lacks the ability to interact with G3BP1/2 proteins, which are RNA-binding proteins that promote stress granule assembly. The RATA mutant virus is attenuated in cell culture models and is less pathogenic in K18-hACE2 transgenic mice. In comparison to WT SARS-CoV-2, G3BP displays reduced co-localization with dsRNA and N at DMV replication factories in the RATA mutant virus infection. The authors also observe less incorporation of puromycin at DMV in the RATA mutant virus infection, suggesting a reduction in local translation of viral RNA at the DMV. Based on these observations, the authors claim that G3BP1/2 proteins are host factors that are required for SARS-CoV-2 replication by enhancing localized translation at viral DMV.

In general, the authors describe interesting cellular biology relating to how SARS-CoV-2 N modulates G3BP functions during infection, which adds to a growing body of literature. The paper is well-written and the data is of high quality.

However, their data do not support their primary conclusion that G3BP is a pro-viral protein required for SARS-CoV-2 replication. Specifically, the observation that SARS-CoV-2 replicates to similar titers in parental and G3BP-KO cells (data not shown), which is consistent with other studies (Burke et al., RNA 2021 PMID: 34315815), indicates that G3BP is not a pro-viral protein that is required for SARS-CoV-2 replication. These data strongly argue against their primary model. An alternative explanation for the attenuation of the RATA virus is that the reduction in interaction between G3BP1 and N during RATA mutant virus infection leads to enhanced G3BP1-mediated antiviral activity (i.e., type I IFN response, as suggested in their model) and thus reduce RATA mutant virus replication. However, the authors do not examine if and how the RATA mutant alters antiviral responses, nor do they test if the replication of the RATA mutant would be rescued in G3BP-KO cells as would be expected by this function of N. Because the RATA mutations of N could disrupt a number of putative N functions, it is unclear if altered G3BP interactions contribute to RATA virus attenuation.

Overall, their data do not support their primary conclusion that G3BP-N interaction is required for optimal viral replication via localization of translation at viral DMV, which weakens the impact of this article.

Specific comments:

1. Extended Figure 3D. Most infected cells do not contain G3BP1 complexes in cells infected with either RATA or WT virus. Most cells with G3BP1 granules do not appear to be infected. Are these SGs generated through paracrine signaling from

infected cells? Also, eIF4G staining is only observed in SARS-CoV-2 infected cells. Does SARS-CoV-2 infection lead to an increase in eIF4G, or is this signal spectral crossover from viral N staining?

2. Burke et al. 2024 (PMID: 38295168) showed that an N-resistant G3BP1 could cause G3BP1 aggregates to form in SARS-CoV-2 infected cells, and that inhibition of eIF4A but not sodium arsenite-induced phosphorylation of eIF2-alpha increased G3BP1 interactions with viral RNA in large aggregates containing viral RNA and dsRNA. Because these findings are similar to those made by the authors in this study, the authors should cite accordingly.

3. Fig. 1A. It is not clear that the cells with SGs are infected since they lack N protein.

4. Fig. 2G. The authors show that the SARS-CoV-2-RATA mutant is attenuated, as it replicates to lower titers in several cell types. While the authors claim that this is the result of disrupting G3BP1-N interactions required for maximal viral replication capacity, it could also be that this mutation disrupts normal N functions or disrupts interactions with other host proteins. If the attenuation of the RATA mutant is due to disruption of G3BP1, then knockout of G3BP1 would be expected to reduce SARS-CoV-2-WT virus replication capacity similarly. The authors should test if SARS-CoV-2 replicates to lower titers in G3BP-KO cells in comparison to parental cells, and if so, show that rescue of G3BP1 rescues viral replication. Notably, Knockout of G3BP1/2-KO did not reduce SARS-CoV-2 replication in A549 cells (Burke et al., 2024).

5. Fig. 2G. If the RATA mutant leads to enhanced antiviral activities of G3BP, then it should replicate to equal titers as WT virus in G3BP-KO cells. The authors should consider testing this.

6. Lines 182-183. "As expected, N-RATA colocalized neither with G3BP1 nor dsRNA (Fig. 5b), demonstrating that SARS-CoV-2 N recruits G3BP1 to RTC". This statement is misleading based on the data in the figure, as N-RATA does co-localize with dsRNA but not with G3BP1.

7. Fig. 6C. It is not obvious that ribosomes are reduced near DMV in the RATA mutant compared to WT virus. The authors should quantify this result.

Version 1:

Reviewer comments:

Reviewer #1

(Remarks to the Author)

The manuscript Long et al. investigates the pro-viral role of G3BP1/2 during SARS-CoV-2 infection. Previous studies have established that the SARS-CoV-2 Nucleocapsid (N) protein binds with G3BP1/2, inhibiting stress granule formation (which is believed to be antiviral). However, whether G3BP1/2 is itself pro- or anti-viral is a matter of some debate. To investigate the importance of N:G3BP1/2 binding, the authors use reverse genetics to generate a recombinant virus with a two amino acid substitution in the N protein (N: I15A; F17A). Compared to infection with wild type SARS-CoV-2, the authors mutant exhibits reduced replication, N:G3BP1/2 binding, and increased stress granule formation.

Compared to the initial submission, this manuscript has been great improved. All of my major concerns were addressed, specifically 1) my concerns regarding novelty, 2) the specificity of the I15A, F17A mutant compared to F17A alone, 3) the lack of replication data in vivo, and 4) data strengthening their claim regarding an effect on translation of viral proteins. In addition, they have adequately addressed all of my minor critiques.

As such, I recommend this article is suitable for publication as is.

Reviewer #2

(Remarks to the Author)

In this revised manuscript, the authors have improved the work. I am supportive of publication but recommend some final alterations be made to the manuscript to strengthen the work and remove ambiguities.

1) An important experiment is how the WT and RATA virus replicate in WT and G3BPΔΔ cells since this addresses whether G3BP proteins are primarily antiviral or proviral and the nature of the alteration in the RATA mutant. I thank the authors for adding these experiments. The work would be strengthened by clarifying the specific differences observed in g3bpΔΔ cells. As I look at the figure I observe:

a) The RATA mutant is 10-50 times less replicative than WT virus in WT cell lines, but only ~3X worse than WT in g3bpΔΔ cells, consistent with the authors interpretation that the major effects of this mutation are due to G3BP proteins.

b) It looks like WT virus is hindered in g3bpΔΔ cells suggesting G3BP proteins can promote replication, but this effect is not rescued by the reintroduction of G3BP1 (assuming I am interpreting the figure correctly).

I suggest: i) The authors clarify what the differences are for WT and RATA virus in the different cell lines and then ii) describe how they interpret those differences for the functional consequences of the N-G3BP interaction.

2) The authors interpretation that N-G3BP promotes viral translation could be strengthened by quantifying the puromycin labeling experiments on a single cell level (using data they already have). Since most of the translation at this stage will be viral mRNAs, it would be predicted that either RATA in WT cells, or WT virus in g3bp $\Delta\Delta$ cells, should show overall reduced translation rates.

3) It would be appropriate to at least discuss the alternative model wherein G3BP plays a role in promoting virion packaging and release from the cells (Murigneux et al., Nature Communications, 2024). Could both models be true, or might there be a simpler resolution?

4) A terrific experiment would be to show immuno-gold localization of G3BP to the DMV with ribosomes on them. I would not require that for publication, but it would really strengthen the work.

Reviewer #3

(Remarks to the Author)

In this revised manuscript, the authors adequately addressed many reviewer concerns. However, their data in Fig. 3G do not support that G3BP1 is pro-viral during SARS-CoV-2 infection. Specifically, the authors show that the RATA virus displays a reduced ability to generate plaque forming units in several cell lines, including Vero cells, MA-104, and U2OS cells. This indicates that the RATA virus is attenuated. However, whether this is due to the inability of N to interact with G3BP1 during RATA infection was unknown.

To address this, the authors examined growth kinetics of WT and RATA via plaque assays in parental (U2OS-ACE2), G3BP1-KO (GFP), and rescue (GFP-G1-WT) U2OS cell lines. Several observations do not support that G3BP1 is a pro-viral host factor required for SARS-CoV-2 replication based on data in Fig. 3G:

1. Knockout of G3BP1 in U2OS cells resulted in higher titers of WT virus by 36 hrs. p.i. Thus, G3BP1 is not required for SARS-CoV-2 replication, but in fact could reduce SARS-CoV-2 replication. Moreover, rescue of GFP-G3BP1 in the G3BP1-KO cells (GFP-G1-WT) did not enhance WT SARS-CoV-2 replication kinetics or in increase final PFU titers. This indicates that G3BP1 is not a host factor that enhances SARS-CoV-2 replication.
2. Knockout of G3BP1 in U2OS cells increased RATA virus replication, suggesting that G3BP1 perturbs RATA virus replication. Notably, RATA virus replication is reduced by GFP-G3BP1 in (GFP-G1-WT) U2OS cell lines.
3. The RATA virus is attenuated in G3BP1-KO (GFP) cells in comparison to WT cells. This indicates that the mutations in N that lead to attenuation of the RATA virus is G3BP1-independent.
4. Despite the overall higher replication of WT vs. RATA in all the cell lines, the growth kinetics between WT and RATA viruses is similar between parental (U2OS-ACE2) and G3BP1-KO (GFP) cells.

In summary, if the interaction between SARS-CoV-2 N and G3BP1 were required to enhance viral replication, then the authors would have observed a reduction in WT virus replication in G3BP1/2-KO cells to titers equivalent to RATA. Moreover, the reduction in WT virus fitness would be rescued in (GFP-G1-WT) U2OS cell lines. However, the authors do not observe these effects. Instead, they observe that knockout of G3BP1/2 leads to higher replication of both WT and RATA, and that rescuing G3BP1 expression only reduces RATA. Combined, these data indicate that the RATA mutations attenuate SARS-CoV-2 replication independently of G3BP1/2 interactions, and that the slower replicating RATA virus might be more sensitive to the general antiviral effects of G3BP1 (interferon-independent).

Additional comments

Considering the observation that their viral N staining is capable of contaminating other channels, the authors should consider repeating key results with no-primary controls to confirm that any co-localization results are not due to spectral crossover.

Version 2:

Reviewer comments:

Reviewer #2

(Remarks to the Author)

The authors have adequately addressed my comments.

Reviewer #3

(Remarks to the Author)

The manuscript is improved from initial submission, and the authors have addressed many reviewer concerns. Overall, this is a thorough study and thus publication is recommended.

REVIEWER COMMENTS

Reviewer #1 (Remarks to the Author):

Summary/Key Results

The manuscript Long et al. investigates the pro-viral role of G3BP1/2 during SARS-CoV-2 infection. Previous studies have established that the SARS-CoV-2 Nucleocapsid (N) protein binds with G3BP1/2, inhibiting stress granule formation (which is believed to be antiviral). However, whether G3BP1/2 is itself pro- or anti-viral is a matter of some debate. To investigate the importance of N:G3BP1/2 binding, the authors use reverse genetics to generate a recombinant virus with a two amino acid substitution in the N protein (N: I15A; F17A). Compared to infection with wild type SARS-CoV-2, the authors mutant exhibits reduced replication, N:G3BP1/2 binding, and increased stress granule formation.

The most interesting findings of this article are mechanistic. The authors clearly demonstrate that G3BP1 co-localizes with NSP3 on double membrane vesicles and that this interaction requires N:G3BP1 binding. Furthermore, they provide evidence that this results in enhanced translation at RTCs, evidenced by the co-localization of 40S ribosomal subunits, ribopuromycylation experiments, and the number of ribosomes localized to DMVs by TEM.

Overall, this manuscript is both interesting and scientifically sound. In particular, the last 1/3 (Fig. 5 and 6) of the paper linking N:G3BP1 binding with enhanced translation at DMVs is novel. Pro-viral functions of G3BP1 during SARS-CoV-2 infection have been proposed, but the data presented here are the first to elucidate a concrete mechanism.

Despite this article's strengths, there are some major weaknesses. First, the overall novelty of the paper is lessened by the fact that a similar article, Yang et al. 2024, has already been published. In Yang et al., the authors produced a single substitution mutant (F17A) SARS-CoV-2, and described the effects on in vitro replication, G3BP1/2 binding, phase separation, stress granule formation, and in vivo infection. Given the similarity in the mutants (F17A vs I15A/F17A) and that the results and conclusions are almost identical (they differ only in magnitude), most of the data presented in Figures 2 – 4 are merely a confirmation of prior work. At minimum, the authors need to cite this past work.

In conclusion, the mechanistic work present in the figures 5 and 6 of this manuscript are novel, interesting, and important. However, this novelty is harmed by the fact that the data in Figures 2 – 4 is largely a confirmation of a manuscript of another group, which the authors do not cite. The authors can improve their manuscript by a) emphasizing and expanding upon those findings which are novel and b) citing Yang et al. and addressing the minor points of difference between the two studies.

We thank the reviewer for the positive comments and valuable suggestions.

Specific Critiques:

Major:

1. Novelty: Yang et al. have already published on a recombinant SARS-CoV-2 mutant that disrupts N:G3BP1/2 binding. Their mutant contains 1 of the 2 mutations used by the authors (F17A) and show similar results. Thus, Yang et al. has essentially already described the following

data included in this manuscript:

- a. structure of the N:G3BP interface within the NTF2L domain (Fig. 2A)
- b. disruption of N:G3BP1 binding through the F17A mutation (Fig. 2C)
- c. increased stress granules in cells infected with the F17A mutant compared to WT (Fig. 2D)
- d. decreased viral replication in vitro in the F17A mutation (Fig. 2G)
- e. decreased pathogenesis and replication in vivo using a small animal model (Fig. 3B, 3C)
- f. That G3BP1 facilitates LLPS, which is disrupted by the F17A mutation (Fig. 4)

While the mutants are not identical (F17A alone in Yang, vs I15A and F17A in the present manuscript) the same conclusions are reached. At minimum, Yang et. al. should be cited. Overall, this reduces the novelty of the research significantly.

The points about the decreased novelty of parts of our paper after the publication of the Yang paper are well taken. It is an area of great interest and indeed, some of those discoveries listed had been made before Yang, including (a) by Biswal *et al.*, *J Mol Biol* 2022, PMID: 35240128 and (b) by Kruse *et al.*, *Nature Communications* 2021; PMID 34799561; Huang *et al.*, 2021, *Cell Discovery*, PMID: 34400613; Biswal *et al.*, *J Mol Biol* 2022). The Yang paper was cited in the original manuscript (reference 23) and we have now cited it more prominently and discussed their data on lines 130-132 in the revised manuscript.

A major difference between the studies is our inclusion of the I15A mutation. Our strategy, based on our previous work on viral FG-based motifs binding to G3BP, was to mutate more than one critical residue to confidently predict that no binding would take place in infected cells.

The NTF2-like domain of G3BP1 features a long binding groove formed by two α -helices and two β -sheets, comprising a 5.6 Å wide groove and a 3.5 Å narrow groove. In our previous structural work, a dual groove-insertion mode was observed in complexes involving alphavirus nsP3/G3BP1-NTF2L (Schulte *et al.*, *Open Biology* 2016, PMID: 27383630) and the host protein Caprin1/G3BP1-NTF2L (Schulte *et al.*, *Open Biology* 2023, PMID: 37161291). Taking N for example in this binding mode, the N-F17 aromatic ring inserts into the aromatic cage at the centre of NTF2L binding groove, stabilised by multiple π -stacking interactions, while the bulky hydrophobic side chain of N-I15 inserts into the small groove, coordinated by G3BP residues L10, V11, and P6. Notably, the Caprin1-Y370A mutation (equivalent to N-I15) significantly reduced Caprin1's binding to G3BP-NTF2L (Schulte *et al.*, *Open Biology* 2016, PMID: 27383630). Therefore, we postulated that both I15 and F17 in N were crucial for its interaction with G3BP-NTF2L and mutating them to alanine could maximally disrupt this interaction.

In the Biswal and Yang papers, the F17A single mutation abolished binding to N in *in vitro* assays, but a low but detectable level of binding was detected in a smaller study by Huang *et al.*, 2021, *Cell Discovery*, PMID: 34400613, see their Fig S1C). We were concerned that in infected cells, where there will be other mechanisms for grouping and concentrating viral and cellular components, weak binding affinities might be compensated by high local concentrations of the relevant proteins.

Indeed, the data presented by Yang *et al.*, in Figure 6E and F, show that there is weak, but readily detectable colocalisation of G3BP and N-F17A at 24 hours of infection in VeroE6-TMPRSS2 cells, (a detail from their Figure 6F reproduced below in Rebuttal Fig 1). Compare those data with the equivalent experiments from our study with RATA (Figs 6a, c, e, 7a, S3a, S5) where there is negligible colocalisation of N-RATA and G3BP1.

Rebuttal Figure 1: Data reproduced from Yang *et al.* Cell Reports, figure 6F with red lines and box added to indicate sites of colocalisation of G3BP1 (yellow), viral genomic RNA (magenta) and N-17A (cyan).

This minimal level of recruitment of G3BP by N-F17A is probably insufficient to counteract the antiviral effect, and the conclusions of Yang and colleagues are unaffected. However, it remains possible that a low level of interaction of G3BP and N-F17A, *earlier* in infection might be able provide some of the proviral effects that might explain the difference in magnitude of the attenuation between F17A (Yang) and I15A+F17A (our work).

These data, combined with the specificity of the I15A mutation for G3BP (see our response to comment 2, below), reinforces our decision to mutate both residues to create viral mutant that is truly defective for G3BP interaction. We believe this adds to the novelty of our work and hope the reviewer agrees.

In a further change to strengthen the novelty of our work relative to that of Yang and colleagues, we have moved data using virus variants of concern, from supplement to the main figures (new Fig 2). We show, similarly to Yang *et al.*, that the P13L mutation in the N protein of the omicron lineage causes a slight decrease in the binding to G3BP. However, we extend that work with infectious WA-1 (P13) and XBB.1.5 (L13) viruses, showing that, despite the slightly decreased binding, no difference in SG inhibition is observed, suggesting it unlikely that this mutation might contribute to reduced virulence of omicron variants, as suggested by Yang *et al.*

But, as the reviewer points out, the real novelty in our work lies in the mechanistic work which is “the first to elucidate a concrete mechanism” for G3BP’s proviral effects. In response to other comments (below), we have further strengthened those experiments and now believe the work represents an advance to the field. We hope the reviewers and editor agree that the paper is now worthy of publication.

2. Specificity of Mutant: While the underlying conclusions of Yang *et al.* are concordant with the authors work, there is a major difference in the magnitude of the effects seen. For example, Yang *et al.* reports ~ 1-log reduction in virus replication with their mutant (F17A) but the present study reports ~2-log reduction with theirs (I15A, F17A). The Yang *et al.* manuscript spends a great deal of time demonstrating that the F17A mutation is highly specific, disrupting SARS2 N:G3BP1/2 binding while not affecting SARS2 N’s affinity to any other cellular proteins according to mass spectrometry analysis. Thus, one possible explanation for the differences in the magnitude of phenotypes between the two studies is that the addition of I15A results in off target effects. Control experiments addressing the specificity of the double mutant are needed to alleviate these concerns.

We do not believe that the I15A mutation results in off target effects. Our main evidence for this is an experiment performed in our collaborator Jakob Nilsson’s lab in Copenhagen, and presented in Kruse *et al.*, Nature Communications 2021; PMID 34799561), figure 2i (reproduced here as Rebuttal Fig 2). The results show that the R14A and I15A double mutant is very specific for G3BP1 and G3BP2. This is in agreement with our observations (mentioned above) that both I15 and F17 are critical for G3BP interaction. Mutation of either residue abolishes G3BP interaction in most experiments, but we chose to mutate both to be sure to exclude any weak interactions.

Rebuttal Figure 2: Reproduced from Kruse et al., 2021, figure 2i legend – “Quantitative mass spectrometry analysis of YFP-tagged SARS-CoV-2 N WT or 2A purified from HeLa cells (n = 4 technical replicates).”

3. Viral replication in K18-hACE2 mice. Fig. 3 compares the pathogenesis of SARS2 WT vs the I15A, F17A mutant, demonstrating that the I15A, F17A mutation reduces weight loss, H&E staining, and antigen staining relative to WT. The absence of viral lung titer is a striking omission. Viral lung titers between SARS2 WT and the I15A, F17A mutation should be examined to confirm replication differences *in vivo*.

We present RT-qPCR data from fixed lung tissue for viral RNA quantification as a measure of viral load in animals from the primary challenge and from the re-challenge. The results, now included in the revised manuscript as Fig 4c and 4f respectively (and reproduced here in Rebuttal Fig 3), clearly show that the RATA mutant is attenuated for replication *in vivo* (primary challenge) and that mice previously infected with RATA are better able to control WT virus infection (rechallenge)

We did not save fresh-frozen lung tissue from which to quantify infectious viral lung titres, but we believe repeating the experiment to generate live viral titres would be unnecessary and not be in accordance with the “3R” guiding principles for more ethical use of animals (*Reduction*).

Rebuttal Figure 3: RT-qPCR of N and E mRNA expression in mice lungs after primary challenge and after rechallenge.

4. Translation of viral proteins: As mentioned in the summary, the authors finding that N re-localizes G3BP1 from stress granules to replication transcription complexes to enhance translation is a major strength and a real step forward for the field. However, when digging into the details of their model, the authors propose that the ribopuromycylation staining co-localizing with NSP3 indicates enhanced translation of viral genes as they emerge from DMVs. While plausible, this is not adequately supported. Looking at Fig. 2C, the protein levels of N itself are not affected much by the I15A, F17A mutation. This should be explained, or the levels of other viral proteins should be examined. Given the normal abundance of N within infected

cells, viral factors translated from longer subgenomic transcripts may be affected more drastically and validate their model.

We thank the reviewer for the positive comments on this aspect of our work and the constructive critique. We have strengthened this aspect of the work in the following ways:

1. We show that total levels of viral proteins at early stages are lower in RATA infected cells (Fig 7d of the revised manuscript and reproduced here as Rebuttal Fig 4). We restricted these analyses to an early time point since differences in RNA replication and transcription confound analyses at later times. At 6 hours post infection, when viral mRNA levels are equivalent (Fig 7c of revised manuscript), we now show that total levels of spike and N protein are lower in RATA infected cells than WT.

Rebuttal Figure 4: VeroE6 cells were infected with SARSCoV-2 WT or RATA mutant at 0.05 MOI for 6h, and cells were lysed for immunoblotting with indicated antibodies. Representative images from three independent experiments are shown. Quantification of western blot was performed using Image J.

2. We have quantified the number of ribosomes in association with DMVs in WT and RATA infected cells. This is better described in our response to Reviewer 2, point 1e. These new data are presented in Fig 7g of the revised manuscript.

Minor:

1. While the microscopy data presented in Fig.5 and Fig 6 (IF and TEM) is convincing, it would be improved by any sort of quantification that the authors would be able to provide. For instance, the number of ribosomes per DMV in Fig. 6C, etc.

We have quantified co-localisations (Pearson's coefficient) in several places in the manuscript and quantified the number of ribosomes per DMV in a revised figure 7 and supplementary figure 6. This latter point is better described in response to Reviewer 2, point 1e.

2. The scale bars on all the IF images throughout the manuscript are too small to read.

We have improved the presentation of microscopy images throughout the manuscript in response to this and similar comments from other reviewers.

Reviewer #2 (Remarks to the Author):

This manuscript addresses the mechanisms by which G3BP proteins affect the ability of SARS2 to infect human cells. The work concludes that the G3BP proteins interact with the SARS2 N protein to promote the translation of viral RNAs that are emerging from the membrane bound replication organelles. Although this is potentially an interesting manuscript, as detailed below, significant additional analyses would be required to support the major conclusions.

This review is from Roy Parker and I would be happy to clarify these comments for the authors if needed.

Specific Comments:

1) The novel conclusion of this work is that a complex of G3BP-N is targeted to the pore complex of double membrane replication organelles to recruit ribosomes to the emerging viral mRNAs. I am not yet convinced of this conclusion given the current data and make the following comments/suggestions.

a) The first argument for this function is that viruses with an N protein mutation that blocking interaction with G3BP proteins showed reduced replication (similar to what was observed in Yang et al., 2024, Cell Reports). However, it remains possible that G3BP inhibits SARS replication and in the absence of N proteins inhibiting G3BP, there is an inhibitory effect on SARS (as suggested at least in part in Burke et al., 2024, Science Advances). I realize the field is split on this issue, but the authors could clarify this issue by examining WT and RATA SARS replication in WT and $g3bp\Delta\Delta$ cells. This would clarify whether G3BP is a host factor, or just limits SARS infection, and how those putative roles are affected by interaction with the N protein. Without clearly demonstrating G3BP is a required host factor it is difficult to argue the N-G3BP protein interaction is required for viral growth.

We do not believe the field needs to be split on the issue of G3BP's role in SARS-CoV-2 (or indeed any) virus infection. We believe that G3BP has *both* antiviral and proviral functions. In the original submission we had not properly cited your Burke et al paper and we have corrected that now.

It was not our intention to state that G3BP is solely or primarily a pro-viral factor for SARS-CoV-2 replication. The proviral function of recruiting translational apparatus to the DMVs may not be a 'required' function, but rather an accessory function that promotes efficient viral gene expression and replication without being critical for viral replication. In contrast, our and others' earlier work showed that G3BP really is critical for chikungunya virus replication (Schulte et al., Open Biology 2016, PMID: 27383630; Kim et al., PLoS Pathogens, 2016, PMID: 27509095; Götte et al., PLoS Pathogens, PMID: 31199850), and viral replication is undetectable in its absence.

We have now performed SARS-CoV-2 WT and RATA viral replication analyses in $\Delta\Delta$ GFP and $\Delta\Delta$ -GFP-G1-WT cells and present the data in Fig 3g of the revised manuscript. These results are better described in response to a major critique of Reviewer 3.

We have added some text in the manuscript (lines 67, and 294-297) to clarify that we believe the proviral effect that we describe is important but not critical for replication.

b) The second argument for this function is that G3BP overlaps with dsRNA and the N protein earlier in infection. This experiment could be improved by i) showing the individual channels (at least in supplement) since it is not possible to assess how significant these overlaps are with

just one small zoom showing individual channels, and ii) some type of quantification at all the time points. I find it notable that at 12 hours it looks like G3BP is excluded from the area of dsRNA, which is similar to what we observed (Burke et al., 2024, Science Advances). How would exclusion of G3BP from replication areas at this time point fit with the proposed model?

The G3BP, dsRNA and N protein colocalisation data are now presented in revised Fig S5a, showing individual channels and colocalisation analyses (Pearson's coefficients). The data now more clearly show that in SARS-CoV-2 WT infection, G3BP1 is recruited to dsRNA foci at early times (3 and 6 hours), but that it is excluded from those sites at later times (12h (and 24h, not shown)).

We believe that the data fit very well with our proposed model. The very high affinity binding of N protein with G3BP will likely mean that most N protein molecules synthesised in the first hours of infection will bind to G3BP, leading to its stoichiometric neutralisation and consequent block in SG formation. The N-G3BP complexes accumulate around the DMVs where the proviral functions of G3BP are carried out. Later, as more N proteins continue to be synthesised, becoming one of the more abundant viral proteins in the cell, the majority of the N staining is then associated with progeny virus particle assembly.

c) The third argument for this function of GFP-G3BP is the overlaps of dsRNA, N, and the Nsp3 protein (as a marker of replication organelles). I make two comments on this experiment. i) I am concerned this staining pattern could be affected by the GFP tag on G3BP1. I suggest this possibility because we observed a different distribution of GFP-G3BP and untagged G3BP in SAR infected cells (see Figure below). Because of this difference, we avoided the use of GFP tagged G3BP for SARS experiments. At a minimum, the authors need to show this subcellular distribution of proteins is not affected by the GFP tag. ii) In addition, however this experiment is performed (e.g. GFP or IF), the experiment could be improved by showing the individual channels (at least in supplement) and quantifying the extent of co-localization.

i) Thank you for sharing your data on (GFP)-G3BP distribution in SARS-CoV-2 infection. We understand the concern and can report that we do not observe any differences in (EGFP)-G3BP1 localisation in parental U2OS or in the U2OS- $\Delta\Delta$ GFP-G1-WT cells after infection with SARS-CoV-2. To illustrate this, in Rebuttal Fig 5 we provide an image of U2OS-ACE2 cells infected with SARS-CoV-2 at MOI 0.5 for 6 hours. Similar to the EGFP-G3BP1 reporter in U2OS- $\Delta\Delta$ GFP-G1-WT cells (see Fig 6c and e of the revised manuscript), endogenous G3BP1 co-localised very strongly with dsRNA and N protein.

Rebuttal Figure 5: U2OS-ACE2 cells were infected with SARS-CoV-2 at 0.5 MOI. Cells were fixed at 6 h and stained for endogenous G3BP1 (green), dsRNA (red) and N (grey).

It is also worth mentioning that we have also used these cells (before ACE2 lentivirus transduction) in our studies with Semliki Forest virus and chikungunya virus subversion of G3BP functions (Panas *et al.*, PLoS Pathogens 2015, PMID: 25658430; Kedersha *et al.*, JCB 2016, PMID: 27022092; Götte *et al.*, PLoS Pathogens, PMID: 31199850, Götte *et al.*, J Virol 2020, PMID:

31941782) and have always found the EGFP-G3BP1 protein to be a faithful reporter for G3BP localisation to viral RNA replication complexes, in comparison with parental U2OS cells and with other cell lines (BHK, Vero, MEF and others).

To further strengthen this part of the work, we have also now performed the G3BP, N, and the nsp3 co-immunoprecipitation experiments using parental U2OS-ACE2 cells and Vero cells (both with endogenous G3BP expression) to compare with the data from U2OS- $\Delta\Delta$ GFP-G1-WT cells (EGFP-G3BP1 transgene expression), as presented in the original submission. In all cases, the results show that (GFP)G3BP-N-nsp3 complexes are formed in WT SARS-CoV-2 infection and that the presence of (GFP)G3BP in those complexes is dependent on the RITFG motif in N protein (disrupted in RATA) - see supplementary fig 5e (U2OS-ACE2 cells), Fig 6f (U2OS- $\Delta\Delta$ GFP-G1-WT cells) and Rebuttal Fig 6 showing the data from Vero cells.

Rebuttal Figure 6: VeroE6 cells were infected with WT virus or RATA mutant at 0.01 MOI for 24 h. Cells were lysed and immunoprecipitated with GFP or N antibody for immunoblotting as indicated.

ii) We have improved the presentation of microscopy images throughout the manuscript in response to this and similar comments from other reviewers. Specifically, we now present the dsRNA, N, and the nsp3 colocalisation data with individual channels in Fig 6e of the revised manuscript.

d) The fourth argument for G3BP promoting local translation is that puromycin labeling overlaps with G3BP IF. It was my understanding that puromycin labeling can no longer be relied upon to identify sites of local translation due to rapid diffusion of released peptides, even when using translation elongation inhibitors in conjunction (Enam et al., 2020, eLife). Given this caveat, additional observations would be needed to make a robust conclusion for G3BP marking the sites of translation. (Can you see ribosome clusters overlapping with G3BP by CLEM or EM with gold labeled antibodies?)

We are aware of the observations presented in the Enam paper, but we do not believe this to be a confounding issue in our experiments. In our case, we are comparing two similar conditions - localised translation at the DMVs in the presence or absence of G3BP. For the “Enam diffusion” to be a confounding issue, it would have to be occurring in the absence of G3BP (both in the KO cells infected with WT virus *and* in WT cells infected with RATA), but not in the presence of G3BP (in WT cells infected with WT virus). We consider this improbable.

Nevertheless, to strengthen this aspect of the work, we have performed the following experiments:

i) Ribopuromycylation experiment with only 2 minutes (reduced from 5 minutes in the original submission) of puromycin labelling to restrict potential diffusion of PMY-labelled peptides. These new data confirm that PMY labelling is strong at SARS-CoV-2 dsRNA+ foci, but only when G3BP is recruited there by N-WT. The data are presented in Fig 7a of the revised manuscript. Additionally, we now include data from analyses of correlations of G3BP1-PMY, or G3BP1-N, (Pearson’s coefficients) and PMY maximum intensities in Fig 7b.

ii) To control for mRNA levels at the time of PMY labelling, we have also included qPCR analyses of viral mRNA levels in the same conditions as the PMY experiment. This shows that WT and RATA viral mRNA levels are equivalent at the time of PMY labelling, excluding that the effect could be due to differences in mRNA template availability. These new data are presented in Fig 7c of the revised manuscript.

iii) We now also show that total levels of viral spike and N proteins are affected by the recruitment of G3BP at early times post infection (6h). The data are presented in Fig 7d of the revised manuscript and are better described in response to Reviewer 1, point 4, above.

iv) We have quantified the number of ribosomes in association with DMVs in the presence and absence of G3BP, described better in response to the following comment.

e) The final argument for this function of G3BP is EM imaging showing differences between WT and RATA SARS infection. The key observations are that in RATA mutants and *g3bpΔΔ* cell lines the ribosomes are more diffuse and less concentrated around the double membrane viral organelles. While a single EM image can be interesting, to make these points robustly, those differences need to be quantified in some manner (beyond just the size of the LVCVs).

Indeed, our apparent reliance on a single image was a weak point in the original submission, and we are grateful for the opportunity to improve this aspect of the work. The originally presented image had in fact been chosen as representative of over 15 images taken. We have now taken more images of U2OS-ACE2 cells infected with WT or RATA mutant virus and performed blinded quantifications of the numbers of ribosomes associated with DMV membranes in those images. Specifically, images of 66 DMVs in WT and 57 in RATA-infected U2OS-ACE2 cells were shuffled and labelled in alphabetical order (by SL, first author) for quantification of ribosomes per DMV and measurement of DMV circumference (by MG, second author). Final numbers were collated and expressed as the number of ribosomes per μm of DMV perimeter. The results show that there is a significant decrease in the number of ribosomes in association with DMVs in RATA infection compared to WT. Similar analyses were performed on images from infected U2OS- $\Delta\Delta$ GFP cells and show that, in the absence of G3BP1/2, the difference in number of ribosomes/DMVs is eliminated. These data are now presented in Fig 7g and S6a of the revised manuscript.

We believe this is strong evidence that the recruitment of G3BP to DMVs by the N protein, leads to more efficient translation of nascent viral mRNAs. Raw data are uploaded as source data file with this submission.

Additional Issues:

2) How do the authors identify the infected cells in Figure 1 a/b since the N protein IF is dim at 6 hours. Is the cell making stress granules at 6 hours infected? This should be clarified.

We believe those cells are infected but are not yet showing strong detectable signal for the N protein. In a new experiment, performed under the same conditions, we could detect dsRNA at earlier stage than N protein. To illustrate this, we include below an image of VeroE6 cells infected with WT SARS-CoV-2 at 0.5 MOI for 6 hours and stained for G3BP1, dsRNA and N (Rebuttal Fig 7). The field captures cells at different early stages of infection. The indicated cell shows clear SGs, but still very low N protein signal, while the cells above left and above right have strong and intermediate N signals respectively but neither contain any SGs.

Rebuttal Figure 7: Vero E6 cells were infected with WT SARS-CoV-2 at 0.5 MOI for 6 hours. Cells were fixed and stained for G3BP1 (green), dsRNA (red), N (grey) and Hoechst (blue).

We have better explained this on line 72 of the revised manuscript.

3) The specific model the authors put forth predicts that in the RATA virus (or in G3BP $\Delta\Delta$ cell lines), the translation efficiency of the viral RNAs would be low early in infection. This could be directly tested by measuring the rate of viral protein production and the levels of viral RNAs at the same time point and comparing WT to RATA.

Please see Reviewer 1, point 4, above for a full description of our new data to address this point.

Reviewer #3 (Remarks to the Author):

In this article, Long and colleagues generate a SARS-CoV-2 mutant (RATA) that lacks the ability to interact with G3BP1/2 proteins, which are RNA-binding proteins that promote stress granule assembly. The RATA mutant virus is attenuated in cell culture models and is less pathogenic in K18-hACE2 transgenic mice. In comparison to WT SARS-CoV-2, G3BP displays reduced co-localization with dsRNA and N at DMV replication factories in the RATA mutant virus infection. The authors also observe less incorporation of puromycin at DMV in the RATA mutant virus infection, suggesting a reduction in local translation of viral RNA at the DMV. Based on these observations, the authors claim that G3BP1/2 proteins are host factors that are required for SARS-CoV-2 replication by enhancing localized translation at viral DMV.

In general, the authors describe interesting cellular biology relating to how SARS-CoV-2 N modulates G3BP functions during infection, which adds to a growing body of literature. The paper is well-written and the data is of high quality.

However, their data do not support their primary conclusion that G3BP is a pro-viral protein required for SARS-CoV-2 replication. Specifically, the observation that SARS-CoV-2 replicates to similar titers in parental and G3BP-KO cells (data not shown), which is consistent with other studies (Burke et al., RNA 2021 PMID: 34315815), indicates that G3BP is not a pro-viral protein that is required for SARS-CoV-2 replication. These data strongly argue against their primary model. An alternative explanation for the attenuation of the RATA virus is that the reduction in interaction between G3BP1 and N during RATA mutant virus infection leads to enhanced G3BP1-mediated antiviral activity (i.e., type I IFN response, as suggested in their model) and thus reduce RATA mutant virus replication. However, the authors do not examine if and how the RATA mutant alters antiviral responses, nor do they test if the replication of the RATA mutant would be rescued in G3BP-KO cells as would be expected by this function of N. Because the RATA mutations of N could disrupt a number of putative N functions, it is unclear if altered G3BP interactions contribute to RATA virus attenuation.

Overall, their data do not support their primary conclusion that G3BP-N interaction is required for optimal viral replication via localization of translation at viral DMV, which weakens the impact of this article.

We thank the reviewer for the comprehensive critique of our work. Below, we provide point-by-point responses to each of the concerns.

Importantly, we would like to stress that we believe that G3BP is both an antiviral and a proviral factor. G3BP-dependent SGs are induced very early in infection, before viral protein production is detectable (Fig 1). Later, when N protein sequesters G3BP to RNA replication complexes, the protein then carries out its proviral functions.

Despite their economical genome coding and preponderance of multifunctional proteins, RNA viruses typically require multiple host factor interactions to carry out functions that cannot be coded for in the relatively small genomes. If one considers the G3BP interaction in this context, it makes sense that the virus sequesters an antiviral protein and uses it for proviral functions.

We address the major critiques in the reviewer's penultimate paragraph:

-“the authors do not examine if and how the RATA mutant alters antiviral responses”

We have now performed qPCR analyses of total mRNA from U2OS-ACE2 cells infected with SARS-CoV-2 WT or RATA for 12 and 24 hours. The data indicate that IFN transcripts are equal (12h) or lower in number (24h) in RATA compared to WT infected cells. We therefore do not believe that the recruitment of G3BP by N protein in WT SARS-CoV-2 infection can be implicated in evasion of the IFN response in infected cells, nor that this could be the molecular mechanism of RATA attenuation. These data are now included in Fig S3e of the revised manuscript.

Further support for this is the observation that the RATA mutant is also attenuated in Vero cells, which lack the ability to produce interferon (Emeny and Morgan, J Gen Virol, 1979, PMID 113494).

-“nor do they test if the replication of the RATA mutant would be rescued in G3BP-KO cells”

In response to the suggestion here, and in points 4 and 5 (below), of testing viral replication in G3BP-KO relative to parental cells, we have now performed WT and RATA viral replication analyses in U2OS- $\Delta\Delta$ GFP and U2OS- $\Delta\Delta$ GFP-G1-WT cells and present those data in Fig 3g of the revised manuscript and in Rebuttal Fig 8 below.

Firstly, to reduce experimental noise caused by unequal expression of EGFP-G3BP and of viral receptor ACE2, we have re-sorted the cell lines based on EGFP expression and on ACE2 expression before performing these analyses. In the interpretation of these data, we would remind the reviewer that we believe that G3BP has both antiviral and proviral effects and, since both are acting on viral replication in such experiment, we would urge caution in drawing conclusions from such a simple readout (infectious viral titre in extracellular medium).

The data show that in U2OS- $\Delta\Delta$ GFP-G1-WT cells, RATA is greatly attenuated relative to WT virus. This we believe is due to both antiviral effects of G3BP SGs and also the lack of the proviral effect of recruiting translational machinery to DMVs. However, in U2OS- $\Delta\Delta$ GFP (lacking G3BP), RATA replicated to titres much closer to WT, due to the absence of the antiviral effect and proviral effects.

Rebuttal Figure 8: U2OS- $\Delta\Delta$ GFP and U2OS- $\Delta\Delta$ GFP-G1-WT cells were sorted for EGFP and ACE2 expression (see methods) and infected with SARS-CoV-2 WT or RATA at 0.05 MOI. Samples were taken at indicated times and titrated by plaque assay on VeroE6 cells.

Specific comments:

1. Extended Figure 3D. Most infected cells do not contain G3BP1 complexes in cells infected with either RATA or WT virus. Most cells with G3BP1 granules do not appear to be infected. Are these SGs generated through paracrine signaling from infected cells? Also, eIF4G staining is only observed in SARS-CoV-2 infected cells. Does SARS-CoV-2 infection lead to an increase in eIF4G, or is this signal spectral crossover from viral N staining?

The reviewer is correct that eIF4G signal in the relevant image was contaminated with N signal. We had not noticed this and are grateful to the reviewer for pointing it out. We believe the problem arose from our use of an old aliquot of the Santa Cruz anti-eIF4G antibody (sc-133155). We have now repeated this experiment using a new batch of the same antibody and present the data in Fig S3a of the revised manuscript.

2. Burke et al. 2024 (PMID: 38295168) showed that an N-resistant G3BP1 could cause G3BP1 aggregates to form in SARS-CoV-2 infected cells, and that inhibition of eIF4A but not sodium arsenite-induced phosphorylation of eIF2-alpha increased G3BP1 interactions with viral RNA in large aggregates containing viral RNA and dsRNA. Because these findings are similar to those made by the authors in this study, the authors should cite accordingly.

The lack of citation of the Burke et al paper was an embarrassing oversight on our part. We have corrected this now (reference 39). Their work is discussed on lines 198-200.

3. Fig. 1A. It is not clear that the cells with SGs are infected since they lack N protein.

We believe those cells are in a very early stage of infection and are not yet showing detectable signal for the N protein. Please see our response to Reviewer 2, point 2 for a fuller explanation.

4. Fig. 2G. The authors show that the SARS-CoV-2-RATA mutant is attenuated, as it replicates to lower titers in several cell types. While the authors claim that this is the result of disrupting G3BP1-N interactions required for maximal viral replication capacity, it could also be that this mutation disrupts normal N functions or disrupts interactions with other host proteins. If the attenuation of the RATA mutant is due to disruption of G3BP1, then knockout of G3BP1 would be expected to reduce SARS-CoV-2-WT virus replication capacity similarly. The authors should test if SARS-CoV-2 replicates to lower titers in G3BP-KO cells in comparison to parental cells, and if so, show that rescue of G3BP1 rescues viral replication. Notably, Knockout of G3BP1/2-KO did not reduce SARS-CoV-2 replication in A549 cells (Burke et al., 2024).

Firstly, we do not believe that the RATA mutation affects interactions with other proteins. This is based on previous work (Kruse *et al.*, Nature Communications 2021; PMID 34799561), where the RATA mutant was originally described (in protein expression constructs, not in replicating virus). In that study, quantitative proteomics from GFP-pulldown experiments (reproduced here in Rebuttal Fig 9, left), revealed that the RATA mutation (there named '2A') was highly specific for G3BP1 and 2 binding.

Rebuttal Figure 9: Quantitative mass spectrometry comparison of YFP tagged SARS-CoV N wt and 2A (RATA) purified from HeLa cells (data reproduced from Kruse et al., 2021, Fig S3c).

Viral replication assays in the G3BP KO cells are shown in Rebuttal Fig 8, above. Of note, the data presented by Burke et al., and mentioned by the reviewer, show viral titres at only one time point (24h post-infection), at which time, replication should be expected to still be in the logarithmic phase and not yet reached plateau (compare with our data in Vero cells in Fig 3).

5. Fig. 2G. If the RATA mutant leads to enhanced antiviral activities of G3BP, then it should replicate to equal titers as WT virus in G3BP-KO cells. The authors should consider testing this.

Viral replication assays presented above, now reveal that indeed the RATA mutant replicates to titres much closer to the WT virus in the U2OS- $\Delta\Delta$ GFP cells.

6. Lines 182-183. “As expected, N-RATA colocalized neither with G3BP1 nor dsRNA (Fig. 5b), demonstrating that SARS-CoV-2 N recruits G3BP1 to RTC”. This statement is misleading based on the data in the figure, as N-RATA does co-localize with dsRNA but not with G3BP1.

We have corrected this mistake (now on lines 200-202) and thank the review for pointing it out.

7. Fig. 6C. It is not obvious that ribosomes are reduced near DMV in the RATA mutant compared to WT virus. The authors should quantify this result.

We have now performed this quantification and present the results in Fig 7g of the revised manuscript. For a fuller discussion, please see our response to Reviewer 2, comment 1e.

REVIEWER COMMENTS

Reviewer #1 (Remarks to the Author):

The manuscript Long et al. investigates the pro-viral role of G3BP1/2 during SARS-CoV-2 infection. Previous studies have established that the SARS-CoV-2 Nucleocapsid (N) protein binds with G3BP1/2, inhibiting stress granule formation (which is believed to be antiviral). However, whether G3BP1/2 is itself pro- or anti-viral is a matter of some debate. To investigate the importance of N:G3BP1/2 binding, the authors use reverse genetics to generate a recombinant virus with a two amino acid substitution in the N protein (N: I15A; F17A). Compared to infection with wild type SARS-CoV-2, the authors mutant exhibits reduced replication, N:G3BP1/2 binding, and increased stress granule formation.

Compared to the initial submission, this manuscript has been great improved. All of my major concerns were addressed, specifically 1) my concerns regarding novelty, 2) the specificity of the I15A, F17A mutant compared to F17A alone, 3) the lack of replication data in vivo, and 4) data strengthening their claim regarding an effect on translation of viral proteins. In addition, they have adequately addressed all of my minor critiques.

As such, I recommend this article is suitable for publication as is.

Again, we thank the reviewer for the comprehensive review of our work and for this positive recommendation.

Reviewer #2 (Remarks to the Author):

In this revised manuscript, the authors have improved the work. I am supportive of publication but recommend some final alterations be made to the manuscript to strengthen the work and remove ambiguities.

1) An important experiment is how the WT and RATA virus replicate in WT and G3BP $\Delta\Delta$ cells since this addresses whether G3BP proteins are primarily antiviral or proviral and the nature of the alteration in the RATA mutant. I thank the authors for adding these experiments. The work would be strengthened by clarifying the specific differences observed in g3bp $\Delta\Delta$ cells. As I look at the figure I observe:

a) The RATA mutant is 10-50 times less replicative than WT virus in WT cell lines, but only ~3X worse than WT in g3bp $\Delta\Delta$ cells, consistent with the authors interpretation that the major effects of this mutation are due to G3BP proteins.

b) It looks like WT virus is hindered in g3bp $\Delta\Delta$ cells suggesting G3BP proteins can promote replication, but this effect is not rescued by the reintroduction of G3BP1 (assuming I am interpreting the figure correctly).

I suggest: i) The authors clarify what the differences are for WT and RATA virus in the different cell lines and then ii) describe how they interpret those differences for the functional consequences of the N-G3BP interaction.

We refer the reviewer to our response to Reviewer 3, below.

2) The authors interpretation that N-G3BP promotes viral translation could be strengthened by quantifying the puromycin labeling experiments on a single cell level (using data they already have). Since most of the translation at this stage will be viral mRNAs, it would be predicted that either RATA in WT cells, or WT virus in g3bp $\Delta\Delta$ cells, should show overall reduced translation rates.

In the revised submission, we had included such analyses of the puromycin labelling data in VeroE6 cells infected with WT or RATA (manuscript Fig 7b and in Rebuttal Fig 1A right, below). Since we have observed that the effect on translation is very localised to the sites of viral RNA replication, we had presented maximum PMY intensity/cell since we believe it to be a more appropriate measure than mean intensity/cell, which averages signal over the whole cell. For the reviewer's interest, we here present the mean intensity analyses of the same data (Rebuttal Fig 1A, left). We observe a slightly lower although non-significant mean signal in RATA infected cells compared to WT.

Furthermore, we have now also similarly analysed the images from WT or RATA infected $\Delta\Delta$ -GFP-G1-WT cells and included those data in a revised Supplementary Fig 7b and presented in here in Rebuttal Fig 1B. Indeed, the results strengthen our interpretation that viral mRNAs are more efficiently translated in WT than RATA.

Rebuttal Figure 1. Mean and maximum intensity of puromycin staining were calculated in CellProfiler for SARS-CoV-2 WT or RATA infected VeroE6 (A) or (B) ΔΔGFP-G1-WT cells.

We also bring to the reviewer's attention, data added in the previous version in response to reviewer 1, point 4. Those data show that whole cell lysates from VeroE6 cells, infected with SARS-CoV-2 WT (MOI 0.05 for 6h) contain more newly produced Spike and N protein than cells infected with the RATA mutant, despite equivalent viral mRNA levels (Fig 7c,d of the revised manuscript). These data also support that WT viral mRNAs are more efficiently translated than RATA.

3) It would be appropriate to at least discuss the alternative model wherein G3BP plays a role in promoting virion packaging and release from the cells (Murigneux et al., Nature Communications, 2024). Could both models be true, or might there be a simpler resolution?

We discussed that on lines 300-305 of the first revised manuscript and depicted it in Figure 8. Indeed, we believe that both models can be true. Murigneux and colleagues propose (last paragraph of their discussion) "a dual functionality of N/G3BP interactions: on the one hand N sequesters G3BP proteins to prevent antiviral SG formation and to circumvent subsequent antiviral immune responses and on the other hand, the virus hijacks the function of G3BP1/2 to favor production of infectious viral particle". Our work supports that and adds the extra function of facilitating viral mRNA translation at the sites of mRNA production, as depicted in Figure 8.

4) A terrific experiment would be to show immuno-gold localization of G3BP to the DMV with ribosomes on them. I would not require that for publication, but it would really strengthen the work.

We agree that this could be very nice to show, but unfortunately, we did not feel that we had the time or resources to do it at this time.

Reviewer #3 (Remarks to the Author):

In this revised manuscript, the authors adequately addressed many reviewer concerns. However, their data in Fig. 3G do not support that G3BP1 is pro-viral during SARS-CoV-2 infection. Specifically, the authors show that the RATA virus displays a reduced ability to generate plaque forming units in several cell lines, including Vero cells, MA-104, and U2OS cells. This indicates that the RATA virus is attenuated. However, whether this is due to the inability of N to interact with G3BP1 during RATA infection was unknown.

To address this, the authors examined growth kinetics of WT and RATA via plaque assays in parental (U2OS-ACE2), G3BP1-KO (Δ GFP), and rescue ($\Delta\Delta$ GFP-G1-WT) U2OS cell lines. Several observations do not support that G3BP1 is a pro-viral host factor required for SARS-CoV-2 replication based on data in Fig. 3G:

1. Knockout of G3BP1 in U2OS cells resulted in higher titers of WT virus by 36 hrs. p.i. Thus, G3BP1 is not required for SARS-CoV-2 replication, but in fact could reduce SARS-CoV-2 replication. Moreover, rescue of GFP-G3BP1 in the G3BP-KO cells ($\Delta\Delta$ GFP-G1-WT) did not enhance WT SARS-CoV-2 replication kinetics or increase final PFU titers. This indicates that G3BP1 is not a host factor that enhances SARS-CoV-2 replication.
2. Knockout of G3BP1 in U2OS cells increased RATA virus replication, suggesting that G3BP1 perturbs RATA virus replication. Notably, RATA virus replication is reduced by GFP-G3BP1 in ($\Delta\Delta$ GFP-G1-WT) U2OS cell lines.
3. The RATA virus is attenuated in G3BP1-KO ($\Delta\Delta$ GFP) cells in comparison to WT cells. This indicates that the mutations in N that lead to attenuation of the RATA virus is G3BP1-independent.
4. Despite the overall higher replication of WT vs. RATA in all the cell lines, the growth kinetics between WT and RATA viruses is similar between parental (U2OS-ACE2) and G3BP1-KO ($\Delta\Delta$ GFP) cells.

In summary, if the interaction between SARS-CoV-2 N and G3BP1 were required to enhance viral replication, then the authors would have observed a reduction in WT virus replication in G3BP1/2-KO cells to titers equivalent to RATA. Moreover, the reduction in WT virus fitness would be rescued in ($\Delta\Delta$ GFP-G1-WT) U2OS cell lines. However, the authors do not observe these effects. Instead, they observe that knockout of G3BP1/2 leads to higher replication of both WT and RATA, and that rescuing G3BP1 expression only reduces RATA. Combined, these data indicate that the RATA mutations attenuate SARS-CoV-2 replication independently of G3BP1/2 interactions, and that the slower replicating RATA virus might be more sensitive to the general antiviral effects of G3BP1 (interferon-independent).

Indeed, we agree with the reviewer that we cannot claim that G3BP is exclusively a proviral factor. As depicted in Figure 8, we believe that G3BP has both antiviral *and* proviral functions. The specific function of G3BP is determined by the proteins or RNA with which it interacts. G3BP is antiviral when it is free to induce SG formation, leading to arrest of viral protein translation. However, SARS-CoV-2 and several other viruses (see citations in the

manuscript text), sequester the protein to their replication complexes in a way that both inhibits the antiviral functions and drives proviral functions at the sites of mRNA transcription and assembly. This view is shared by Murigneux and colleagues (Nature Communications, 2024 PMID 38245532), quoted above, and our work adds the proviral effect of efficient translation of WT viral mRNAs.

We have extensively edited the text of the paper, including title and abstract to better express the view that G3BP is both antiviral and proviral.

In our first revision, we requested caution in the analyses of viral growth replication data in these cell lines and we repeat that here. We believe it is not appropriate to compare viral replication curves *across the different cell lines* but rather it is better to compare WT and RATA virus replication in each cell line separately. The CRISPR knock out cells and GFP-(G3BP1) reconstitutions were generated 10 years ago, and the cells have been passaged independently since then. In addition, in 2021, all 3 cell lines (U2OS parental, $\Delta\Delta$ GFP and $\Delta\Delta$ G3BP1-GFP) were transduced with ACE2-TMPRSS2-expressing lentiviruses, placed under selection and passaged independently. We have tried to minimise variation caused by unequal expression of these transgenes, but the possibility remains that the viruses might be better able to enter and initiate infection in one cell line than the others.

Comparing the two viruses, we observe that RATA exhibits 33-fold lower replication compared to WT in U2OS parental cells, 25-fold lower replication compared to WT in $\Delta\Delta$ GFP-G1-WT cells, but only a 4-fold reduction compared to WT in cells lacking G3BP1/2 ($\Delta\Delta$ GFP cells). We believe therefore that the attenuation is largely G3BP1-dependent and is a result of the combination of the antiviral effects of SGs and the loss of G3BP's proviral effects. However, as the reviewer points out (point #3), the 4-fold reduction in $\Delta\Delta$ GFP cells suggests some G3BP-independent attenuation of RATA, which we cannot exclude. We discuss this on lines 144-149 of the revised manuscript.

We do not believe that RATA is a “slower replicating” virus, since viral RNA levels were equal at 6hpi (Fig 7c), but that the major attenuation effects are seen downstream of the translation of viral mRNAs. Further, we have been careful to define the specificity of the RATA mutation for G3BP (see comment to Reviewer 1, point 2 in first revision) and to show that the mutation does not alter RNA-binding and LLPS properties (figure 5), as well as the binding to other viral structural proteins for assembly (Nature Communications, 2021, PMID: 34799561). Of note, a recent study found that point mutations in the N protein's intrinsically disordered regions (IDRs) can have “nonlocal impact and modulate thermodynamic stability, secondary structure, protein oligomeric state, particle formation, and liquid-liquid phase separation” (Nguyen et al Elife 2024; PMID: 38941236), a possibility we now mention in the text.

Additional comments

Considering the observation that their viral N staining is capable of contaminating other channels, the authors should consider repeating key results with no-primary controls to confirm that any co-localization results are not due to spectral crossover.

To respond to the reviewer's suggestion, we here present images of U2OS-ACE2 cells infected with SARS-CoV-2 WT or RATA and stained with a combination of antibodies against G3BP1, dsRNA and N protein (Rebuttal Fig 2, top left), or the same lacking either G3BP1 (bottom left), dsRNA (top right) or N (bottom right). The data show that, using the same antibodies, techniques and hardware used in the paper, no spectral crossover was detected.

We are now confident that the fluorescence signals in all images are uncontaminated by other channels. For example, to illustrate that N signal is not contaminating other channels, one might examine images from the manuscript of RATA infected cells, where N (Alexa Fluor 647) does not co-stain with G3BP1 (Alexa Fluor 488; Fig 6a), GFP/eIF4A (GFP or Alexa Fluor 568; Fig 6c), or GFP (Fig 6e).

Rebuttal Figure 2. U2OS-ACE2 cells were infected with SARS-CoV-2 WT MOI 0.5 and cells were fixed and stained with the indicated antibody combinations at 6 hours post infection.